# Bacterial effectors mediate kinase reprogramming through mimicry of conserved eukaryotic motifs

Ioanna Panagi[1], Janina H Muench [ID][2], Alexi Ronneau[1], Ines Diaz-del-Olmo [ID][1,3,4], Agnel Aliyath[1], Xiu-Jun Yu [ID][1], Hazel Mak[1], Enkai Jin[1], Jingkun Zeng[1], Diego Esposito [ID][2], Elliott Jennings[1], Timesh D Pillay [ID][1,2], Regina A Günster[1], Sarah L Maslen[5], Katrin Rittinger [ID][2✉] & Teresa L M Thurston [ID][1,3,4✉]

## Abstract

**Bacteria have evolved numerous biochemical processes that underpin their biology and pathogenesis. The small, non-enzymatic bacterial (*Salmonella*) effector SteE mediates kinase reprogramming, whereby the canonical serine/threonine host kinase GSK3 gains tyrosine-directed activity towards neosubstrates, promoting *Salmonella* virulence. Yet, both the mechanism behind the switch in GSK3's activity and the diversity of this phenomenon remain to be determined. Here we show that kinase reprogramming of GSK3 is mediated by putative homologues from diverse Gram-negative pathogens. Next, we identify both the molecular basis of how SteE targets GSK3 and uncover that the SteE-induced tyrosine activity conferred on GSK3 requires an L/xGxP motif. This motif, found in several CMGC kinases that undergo auto-tyrosine phosphorylation, was previously shown to mediate GSK3 autophosphorylation on a tyrosine. Together, we suggest that the SteE family of intrinsically disordered proteins mediates kinase reprogramming via several short linear motifs that each appear to mimic eukaryotic signalling motifs. With this insight comes the potential for the rationale design of synthetic reprogramming proteins.**

**Keywords** Host–Pathogen Interactions; Intrinsic Disordered Proteins; Kinase Reprogramming; Short Linear Interaction Motifs; *Salmonella*
**Subject Categories** Microbiology, Virology & Host Pathogen Interaction; Post-translational Modifications & Proteolysis; Signal Transduction

## Introduction

The pathogenesis of many Gram-negative bacterial pathogens, including *S.* Typhimurium, relies on the delivery of virulence factors termed effectors, through type 3 secretion systems (T3SS) spanning bacterial and host cell membranes. The delivery of effectors enables the precise regulation of diverse host cellular processes, including metabolism, cell cycle progression, as well as adaptive and innate immune signalling (Jennings et al, 2017). Many effectors are enzymes, including the SseK family of glycosyltransferases which modify death domains, as well as the GtgA family of zinc metalloproteases (Pillay et al, 2023; Jennings et al, 2017). A subset of effectors with non-enzymatic activity, such as SteD (Alix et al, 2020) and SteE (Panagi et al, 2020; Gibbs et al, 2020), function as adaptors, often bridging a eukaryotic enzyme toward a non-canonical target.

SteE is a small 157 amino acid protein with no known enzymatic activity or recognisable domains. Instead, following bacterial translocation, SteE interacts with the eukaryotic serine/threonine (S/T) kinase—glycogen synthase kinase 3 alpha/beta (GSK3α/β) (Gibbs et al, 2020; Panagi et al, 2020). GSK3 is constitutively active, yet its cellular activity and substrate selectivity are carefully controlled through multi-layered regulatory mechanisms that include an activatory tyrosine autophosphorylation event. Tyrosine autophosphorylation is reported as a chaperone-dependent, transient process that occurs co-translationally and requires a short linear motif defined as L/xGxP (Cole et al, 2004; Lochhead et al, 2006; Lee et al, 2020). Despite tyrosine autophosphorylation, in homoeostatic conditions, GSK3 strictly modifies substrates on serine or threonine residues, with a preference to phosphorylate a residue 4 amino acids N-terminal of a pre-phosphorylated residue —S/Txxx(p)S/TP (Fiol et al, 1987; Ter Haar et al, 2001). It is therefore of great interest that GSK3 phosphorylates SteE not only at serine and threonine sites (T91 and S141) but also on Y143 (Gibbs et al, 2020; Panagi et al, 2020).

Phosphorylation of SteE on Y143 forms a classical pYxxQ motif in the C-terminal region of SteE. This motif enables recruitment of STAT3 via its Src Homology 2 (SH2) domain and is required for virulence in vivo (Gibbs et al, 2020). Of note, Y143 of SteE is equivalent to residue Y167 analysed in Gibbs et al (Gibbs et al, 2020), with the difference in amino acid numbering arising from a miss-annotation of the start methionine of STM2585 (SteE/SarA) (Panagi et al, 2020). Within this complex, GSK3 phosphorylates a second tyrosine residue, this time Y705 of STAT3. Phosphorylation at this site of STAT3, which is normally mediated by tyrosine kinases such as Janus Kinases (JAKs), causes its dimerisation and

[1]Department of Infectious Disease, Centre for Bacterial Resistance Biology, Imperial College London, London SW7 2AZ, UK. [2]Molecular Structure of Cell Signalling Laboratory, The Francis Crick Institute, 1 Midland Road, London NW1 1AT, UK. [3]Sir William Dunn School of Pathology, University of Oxford, South Parks Road, Oxford OX1 3RE, UK. [4]Bacterial Pathogenesis and Immune Signalling Laboratory, The Francis Crick Institute, 1 Midland Road, London NW1 1AT, UK. [5]Proteomics Science Technology Platform, The Francis Crick Institute, 1 Midland Road, London NW1 1AT, UK. ✉E-mail: katrin.rittinger@crick.ac.uk; Teresa.thurston@path.ox.ac.uk

translocation to the nucleus (Panagi et al, 2020). In the case of *S.* Typhimurium infection, STAT3 phosphorylation drives the initiation of an anti-inflammatory transcriptional programme within the infected macrophage, which governs pathogen survival and persistence (Panagi et al, 2020; Pham et al, 2020; Stapels et al, 2018; Gibbs et al, 2020; Gaggioli et al, 2025). As the molecular details mediating the reprogramming of the amino acid and substrate specificity of GSK3 are lacking, it is unknown whether manipulation of a host kinase specificity represents a virulence mechanism unique to *S.* Typhimurium or a more universal theme in host–pathogen interactions.

Our work reveals the molecular basis for SteE targeting GSK3, defining a putative interface between these two proteins. SteE is one of several proteins within a newly defined family that co-opts GSK3. Through biophysical experiments and the analysis of conserved amino acids, we identify several motifs that underpin the basis of SteE's kinase reprogramming activity, allowing GSK3, which is a canonical serine/threonine kinase to additionally phosphorylate tyrosine residues on neosubstrates. Our step-by-step breakdown on the contribution of each motif to the manipulation of this host kinase lays the foundation for not only the search of eukaryotic kinase reprogramming proteins but also the development of synthetic reprogramming poly-peptides or other molecules.

## Results and discussion

### Kinase reprogramming is functionally conserved in diverse bacterial pathogens

To date, the *Salmonella* virulence factor SteE is the only known protein that can change the amino acid and substrate specificity of a eukaryotic serine/threonine kinase, enabling it to phosphorylate tyrosine residues on a neosubstrate (Panagi et al, 2020). To determine if putative homologues are present beyond *Salmonella* spp. a BLASTp query (Altschul et al, 1990) of the 157 amino acid sequence of SteE was conducted in the NCBI database. Of the identified proteins, the amino acid sequence from *Escherichia coli* strain 187571 serovar 025:H17 was identical to SteE but truncated at residue 138. In contrast, representative putative homologues from several Gram-negative bacteria—*Erwinia pyrifoliae*, *Edwardsiella ictaluri*, *Arsenophonus nasoniae* and *Pseudomonas umsongensis* each displayed a much lower percentage protein sequence identity (Fig. 1A). KML20850 also represents a putative homologue of SteE from two strains of *Leclercia adecarboxylata*, however the source organism is currently unverified (Genome assembly ASM104363v1 and ASM103892v1).

The genome of the plant associated pathogen *Erwinia pyrifoliae*, most notable for causing necrotic disease in Asian pear and strawberry plants (Kim et al, 1999), contains two distinct T3SSs: a conserved *hrp1* gene cluster with associated effectors and enzymes involved in systemic virulence and a second T3SS that is related to the SPI-1 encoded *inv/spa* cluster of *Salmonella* Typhimurium (Oh et al, 2005; Smits et al, 2010). *Edwardsiella ictaluri* represents a human and fish pathogen that causes enteric septicaemia. It is prevalent worldwide in freshwater and marine aquatic animals and as outbreaks cause up to 100% mortality, mass death of infected fish threatens food security and undermines sustainability

(Machimbirike et al, 2022). The genome encodes for a Ssa-Esc like T3SS that resembles the SPI-2 encoded *Salmonella* T3SS and this injectisome is essential for intracellular replication and virulence of the pathogen (Thune et al, 2007; Dubytska et al, 2016). *Arsenophonus nasoniae* is a heritable microbe associated with male-killing in parasitic wasps across Europe (Nadal-Jimenez et al, 2023). These bacteria encode two complete T3SSs, with the second resembling the SPI-1 encoded *inv/spa*-like apparatus of *Salmonella*. Furthermore, open reading frames, resembling at least 10 effectors from *Salmonella* or other Gram-negative bacteria have been identified in the *A. nasoniae* genome (Wilkes et al, 2010; Siozios et al, 2024). Effector homologues have also been identified in other pathogens including a homologue of SopA in *Erwinia* (Kube et al, 2010) and a homologue of SpvC in *E. ictaluri*, which is required for virulence (Dubytska et al, 2022). In contrast, very little is known about *Pseudomonas umsongensis*, except that it was isolated from the soil in the Umsong region of Korea and can convert polyethylene terephthalate (PET)-derived terephthalic acid into a form of biodegradable plastic (Narancic et al, 2021). Altogether, the presence of both a T3SS injectisome and a putative SteE homologue in these bacteria raises the intriguing possibility that kinase reprogramming represents a more general virulence mechanism.

As SteE-induced STAT3 phosphorylation is mediated by the host serine/threonine kinase GSK3, and SteE and GSK3 are found in a complex together (Panagi et al, 2020), we first tested whether representative putative homologues interacted with human GSK3. When expressed and enriched from 293ET cells, GFP-tagged EPC06920, ALT06054 and SAMN04490206 each formed a stable interaction with endogenous GSK3, whereas KML20850 and ArsFIN19530 did not (Fig. 1B). In addition, as observed for SteE, homologues that stably interacted with GSK3, also became tyrosine phosphorylated (Figs. 1B and EV1A). Functional analysis further revealed that exogenous expression of GFP-tagged EPC06920, ALT06054, and SAMN04490206 in human 293ET cells resulted in phosphorylation of endogenous STAT3 on Y705 whereas the expression of KML20850 and ArsFIN19530 did not (Fig. 1C).

Together, we conclude that reprogramming of a host kinase is a biochemical activity associated with proteins expressed beyond *Salmonella*. The capacity of the putative SteE homologues to induce phosphorylation of human STAT3 correlated with their ability to interact with GSK3 and become tyrosine phosphorylated themselves. It is however noteworthy that the reported native hosts of these bacterial pathogens are divergent. We speculate that structural conservation of GSK3 across host species might explain the ability of the putative SteE homologues to manipulate human GSK3. Furthermore, STAT homologues with conserved SH2 domains are found in common hosts of *Edwardsiella ictaluri* such as catfish (Jin et al, 2018). *Edwardsiella ictaluri* infects macrophages and retains a functional T3SS (Rogge and Thune, 2011) but it remains to be determined if STAT phosphorylation occurs upon infection with *Edwardsiella ictaluri* and whether this is important for disease manifestation. In plants, STAT-type linker-SH2 domain factors have been identified (Gao et al, 2004), revealing that the linker-SH2 domain of the STAT transcription factors are one of the most ancient functional domains. Our work now suggests that diverse bacterial pathogens hijack phosphotyrosine signal transduction through kinase reprogramming.

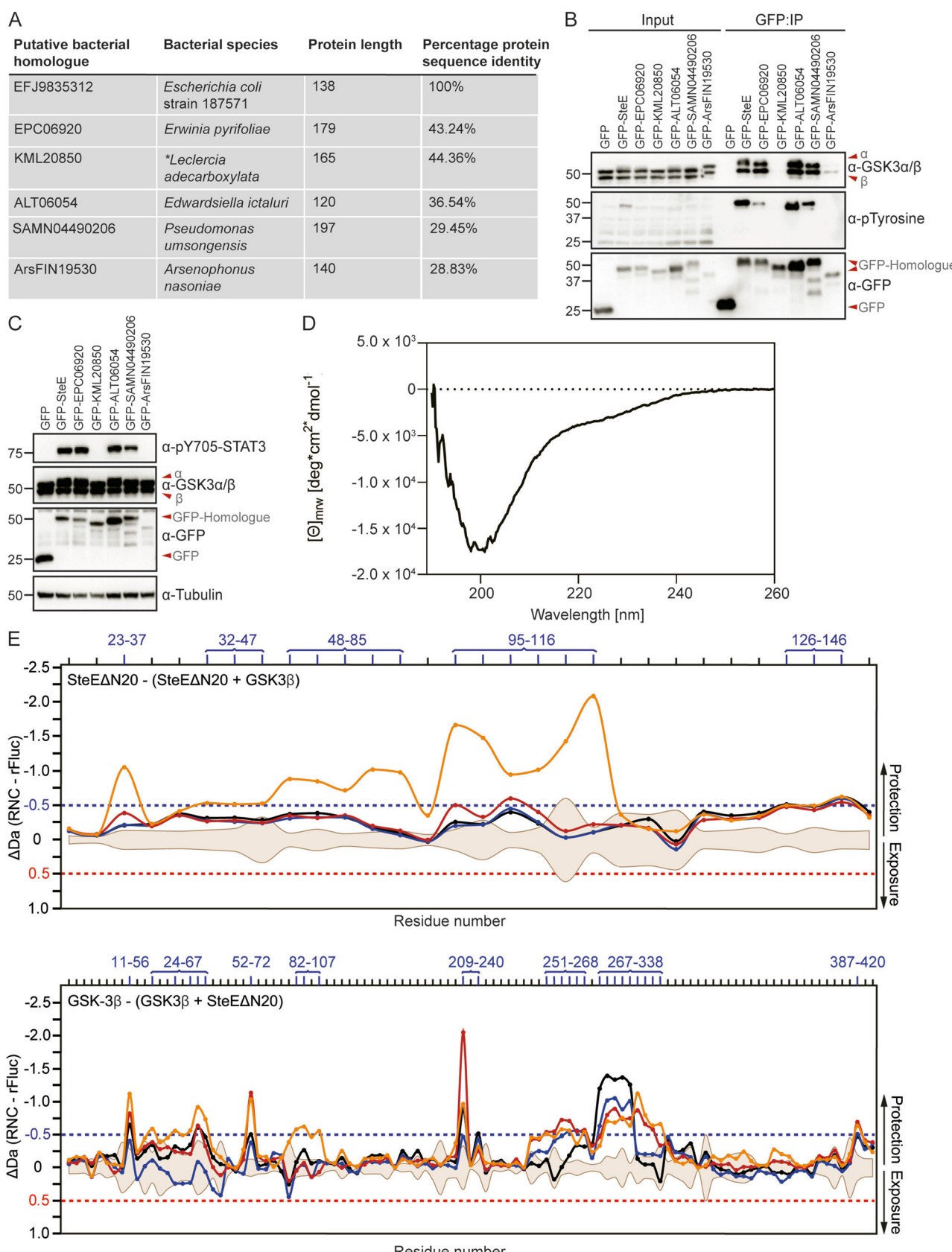

**A**

| Putative bacterial homologue | Bacterial species | Protein length | Percentage protein sequence identity |
|---|---|---|---|
| EFJ9835312 | *Escherichia coli* strain 187571 | 138 | 100% |
| EPC06920 | *Erwinia pyrifoliae* | 179 | 43.24% |
| KML20850 | *\*Leclercia adecarboxylata* | 165 | 44.36% |
| ALT06054 | *Edwardsiella ictaluri* | 120 | 36.54% |
| SAMN04490206 | *Pseudomonas umsongensis* | 197 | 29.45% |
| ArsFIN19530 | *Arsenophonus nasoniae* | 140 | 28.83% |

**SteE**

¹MFTINSTNRV ASTIAPYACV SDVNLEDKAT FLDEHTSIHA NDSSLQCFVL NDQHVPQNTL ATDVEGYNRG LQERISLEYQ
⁸¹PLESIVFLLG TPAVLETKES LSLPVSPDAL TQKLLSISSN DECKLSGSTS CTTPASHNPP SGYIAQYRHS AEVFPDE

Figure 1. Kinase reprogramming is functionally conserved in diverse bacterial pathogens.

(A) Putative homologues of SteE from divergent bacteria as identified with BLAST. The bacterial species, protein length and percentage (%) sequence identity to SteE of each putative homologue are listed as calculated from Clustal Omega alignments to WT SteE (Sievers and Higgins, 2018). The amino acid sequences of SteE and each putative homologue are listed in Appendix Table S3. *KML20850 was originally identified to be present in two strains of *Leclercia adecarboxylata* however the source organism is currently unverified (Genome assembly ASM104363v1 and ASM103892v1). (B) 293ET cells expressing GFP or GFP-tagged SteE homologues were lysed and subjected to GFP-TRAP immunoprecipitation. Samples were analysed by immunoblotting for interaction to endogenous GSK3 and the tyrosine phosphorylation status of each GFP-tagged protein. The data represent two independent biological repeats. (C) Immunoblot analysis of cell lysates obtained from 293ET cells expressing the indicated GFP-tagged SteE homologues. Data represent three independent biological repeats. (D) Far-UV Circular Dichroism spectrum of recombinant His$_6$-SteE$^{\Delta N20}$. (E) Relative hydrogen/deuterium exchange in SteE$^{\Delta N20}$ incubated with GSK3β. Shown are the difference plots of SteE$^{\Delta N20}$ in complex with GSK3β compared to proteins on their own. Aliquots of each sample were treated with D$_2$O for 3 s (orange line), 30 s (red line), 300 s (blue line), and 3000 s (black lines), respectively. Significance is indicated by the dotted red and blue lines. Source data are available online for this figure.

## SteE represents a predominantly unstructured protein

Given that reprogramming of human GSK3 by diverse bacterial proteins was observed, we sought to identify the molecular basis for this unusual mechanism taking SteE as a prototypical family member. Structural models of SteE, generated by AlphaFold3, are of low confidence and suggest a largely unstructured protein (Fig. EV1B). To experimentally investigate the presence of secondary structure within SteE, His$_6$-SteE$^{\Delta N20}$, which lacks the anticipated secretion signal but remains functional (Panagi et al, 2020), was recombinantly expressed and analysed by nuclear magnetic resonance (NMR) spectrometry. 1D and 2D NMR experiments of His$_6$-SteE$^{\Delta N20}$ revealed a low signal dispersion with peaks located between δ 7.8–8.5 ppm which is indicative of a disordered protein that lacks a stable hydrophobic core (Fig. EV1C,D). In addition, analysis by circular dichroism (CD) spectroscopy showed a minimum mean residue ellipticity at 200 nm, which is characteristic of largely unstructured proteins (Fig. 1D). Deconvolution using the DichroWeb server suggests that His$_6$-SteE$^{\Delta N20}$ contains ~65% random coils along with some structured β-strands, β-turns (15% and 11%, respectively) and α-helices (8%). Therefore, in solution, unbound SteE lacks a globular structure.

## Hydrogen/deuterium-exchange mass spectrometry reveals a putative interface between SteE and GSK3

As interaction between SteE and GSK3 is a prerequisite for kinase reprogramming, we sought to identify what defines their protein-protein interface. However, intrinsically disordered proteins, like SteE, are not easily accessible by structural techniques such as X-ray crystallography, and we therefore used hydrogen/deuterium-exchange mass spectrometry (HDX-MS) to identify regions of SteE that are at the interface with GSK3β. HDX-MS reports on the local environment and solvent accessibility of the protein backbone amides and is hence ideally suited to detect protein regions that become exposed or protected upon complex formation. Size-exclusion chromatography demonstrated that recombinant SteE$^{\Delta N20}$ and GSK3 formed a stable complex in vitro, even in the absence of SteE phosphorylation, suggesting that the protein complex is accessible to HDX-MS analysis (Fig. EV1E). Complex formation between GSK3β and SteE$^{\Delta N20}$ resulted in a reduction of deuterium exchange of peptides covering amino acids 23–37, 48–85 and 95–116 of SteE, suggesting that these areas become protected upon binding GSK3β (Figs. 1E and EV2). Concomitantly, analysis of GSK3β revealed several regions that become protected upon

interaction with SteE, with the most prominent protection observed for peptides covering amino acids 209–240 and 267–338 (Fig. 1E).

## Short linear interaction motifs enable SteE to reprogramme GSK3

Our HDX-MS data suggest that multiple regions are involved in the interaction between SteE and GSK3 and that, in particular, amino acids 95–116 of SteE, participate in complex formation. We next investigated whether this region of SteE contains Short Linear interaction Motifs (SLiMs). SLiMs represent stretches of 3–10 amino acids that act as functional modules in intrinsically disordered proteins and the evolution of eukaryotic-like SLiMs in secreted viral and bacterial proteins provides a means by which many pathogens co-opt eukaryotic signalling pathways (Van Roey et al, 2014; Via et al, 2015; Chemes et al, 2015). Sequence alignment between SteE and putative homologues revealed several evolutionarily conserved amino acids, spaced as short motifs, within residues 95–116 of SteE (Fig. EV3). To investigate whether any of these motifs represent SLiMs that might determine the molecular basis for the selective interaction between SteE and GSK3, we analysed SteE variants (SteE$^{KESL-AAAA}$, SteE$^{SLSL-AAAA}$, SteE$^{SLPV-AAAA}$ SteE$^{QKL-AAA}$) that were mutated at select conserved amino acids (Fig. 2A). SteE variants with alanine substitutions were transiently expressed as GFP-tagged fusion proteins in 293ET cells and analysed for their ability to form a stable interaction with GSK3 after their enrichment using GFP-TRAP beads from cell lysates. Wild-type SteE interacted with GSK3 and SteE$^{KESL-AAAA}$ and SteE$^{QKL-AAA}$ retained partial interaction with GSK3. In contrast, SteE$^{SLSL-AAAA}$, and SteE$^{SLPV-AAAA}$, did not form a stable interaction with GSK3 (Fig. 2B). Furthermore, all these mutated forms of SteE were either completely inhibited (SteE$^{SLSL-AAAA}$, SteE$^{SLPV-AAAA}$ and SteE$^{QKL-AAA}$) or severely reduced (SteE$^{KESL-AAAA}$) in their ability to form a complex with STAT3 (Fig. 2B). In line with the findings that SteE$^{KESL-AAAA}$ retained some interaction with both GSK3 and STAT3, when delivered by *Salmonella*, this variant induced Y705 phosphorylation of STAT3 whereas translocated SteE$^{SLSL-AAAA}$, SteE$^{SLPV-AAAA}$ and SteE$^{QKL-AAA}$ failed to do so (Fig. 2C).

Analysis of the amino acid sequence around the SLSLPV motif led us to note that for the putative homologues KML20850 and ArsFIN19530, the amino acid sequence immediately after this motif diverged (Fig. 2A). In SteE, the sequence SLSLPV is followed by an SP to make SLSLPVSP$_{107}$ where S106 is phosphorylated by an unknown kinase (Panagi et al, 2020). This resembles a canonical GSK3 phosphorylation site, defined as SxxxpS(P) (Fiol et al, 1987; Ter Haar et al, 2001). To test the hypothesis that it is the divergence of "SP" in the newly defined SLSLPVSP motif that prevents the

## A

```
STM2585       93-AVLETKESLSLPVSPDALTQKLLS-116
KML20850     125-EALDANESLSLPIFSHVLTQKLLN-148
EPC06920     122-EVLEPDESLSLPASPHILAQKLSS-145
ALT06054      55-EELEDTESLSLAVSPKALHQAISC-78
ArsFIN19530  100-KNIHSAESLSLPVCPFLLSQKLIQ-123
SAMN04490206 144-ERLGIHHQLSLPVSPSMLTEKLIE-167
                 :  ..*** *  :  :
```

| Mutated motif | Amino acid No in SteE |
|---|---|
| KESL to AAAA | K98, E99, S100, L101 |
| SLSL to AAAA | S100, L101, S102, L103 |
| SLPV to AAAA | S102, L103, P104, V105 |
| QKL to AAA | Q112, K113, L114 |

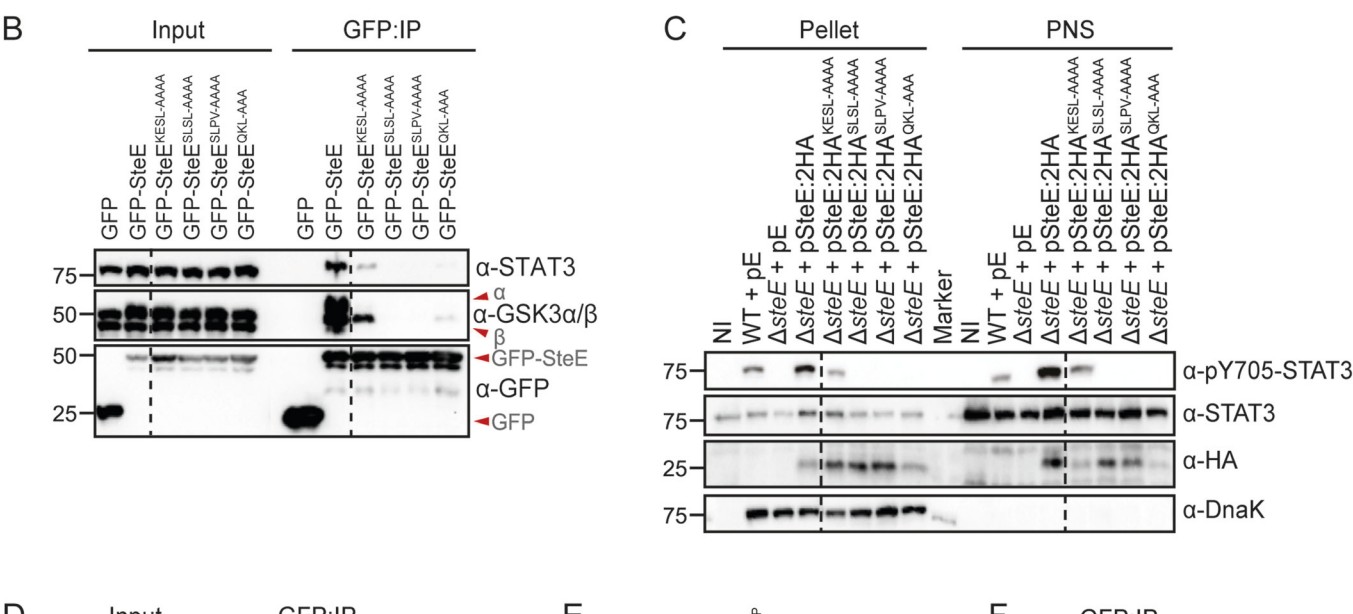

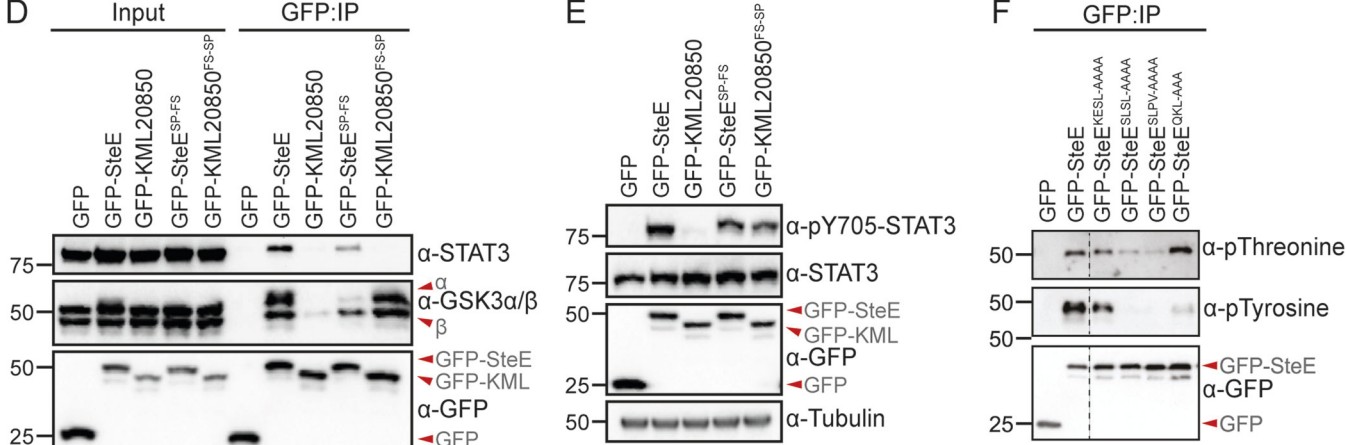

**Figure 2. Several conserved motifs mediate SteE function.**

(A) Sequence alignment between SteE and its putative homologues in the putative GSK3-interacting surface of SteE identified in Fig. 1E. Asterisk (*) = fully conserved residue, Colon (:) = amino acid residues with similar properties, Period (.) = amino acid with weakly similar properties. The underlined region represents the consensus GSK3-recognition motif in SteE. Motifs appearing similar across putative homologues that were mutated in SteE for analysis in subsequent figures are annotated on the right. (B) 293ET cells expressing GFP or the indicated GFP-tagged SteE variant were lysed, and post-nuclear supernatants were subjected to GFP-TRAP immunoprecipitation (IP), followed by immunoblot analysis for interaction with endogenous GSK3 and STAT3. Data represent three independent biological repeats. (C) Cells infected with the indicated *Salmonella* strains for 17 h were lysed, with pellets and post-nuclear supernatants (PNS) separated by centrifugation. Samples were analysed by immunoblotting. Data represent three independent biological repeats. NI non-infected, pE empty vector. (D) 293ET cells expressing GFP or the indicated GFP-tagged proteins were lysed, and post-nuclear supernatants were subjected to GFP-TRAP immunoprecipitation (IP), followed by immunoblot analysis for interaction to endogenous GSK3 and STAT3. Data represent three independent biological repeats. (E) Immunoblot analysis of cell lysates obtained from 293ET cells expressing GFP or the indicated GFP-tagged proteins. Data represent three independent biological repeats. (F) 293ET cells expressing GFP or the indicated GFP-tagged proteins were lysed, and post-nuclear supernatants were subjected to GFP-TRAP immunoprecipitation (IP). Samples were analysed by immunoblotting for the threonine and tyrosine phosphorylation status of each GFP-tagged protein. Data represent two (pThreonine blot) or three (pTyrosine blot) independent biological repeats. Dotted lines on immunoblots represent lanes removed from the analysis, with all samples shown from one continuous membrane. Source data are available online for this figure.

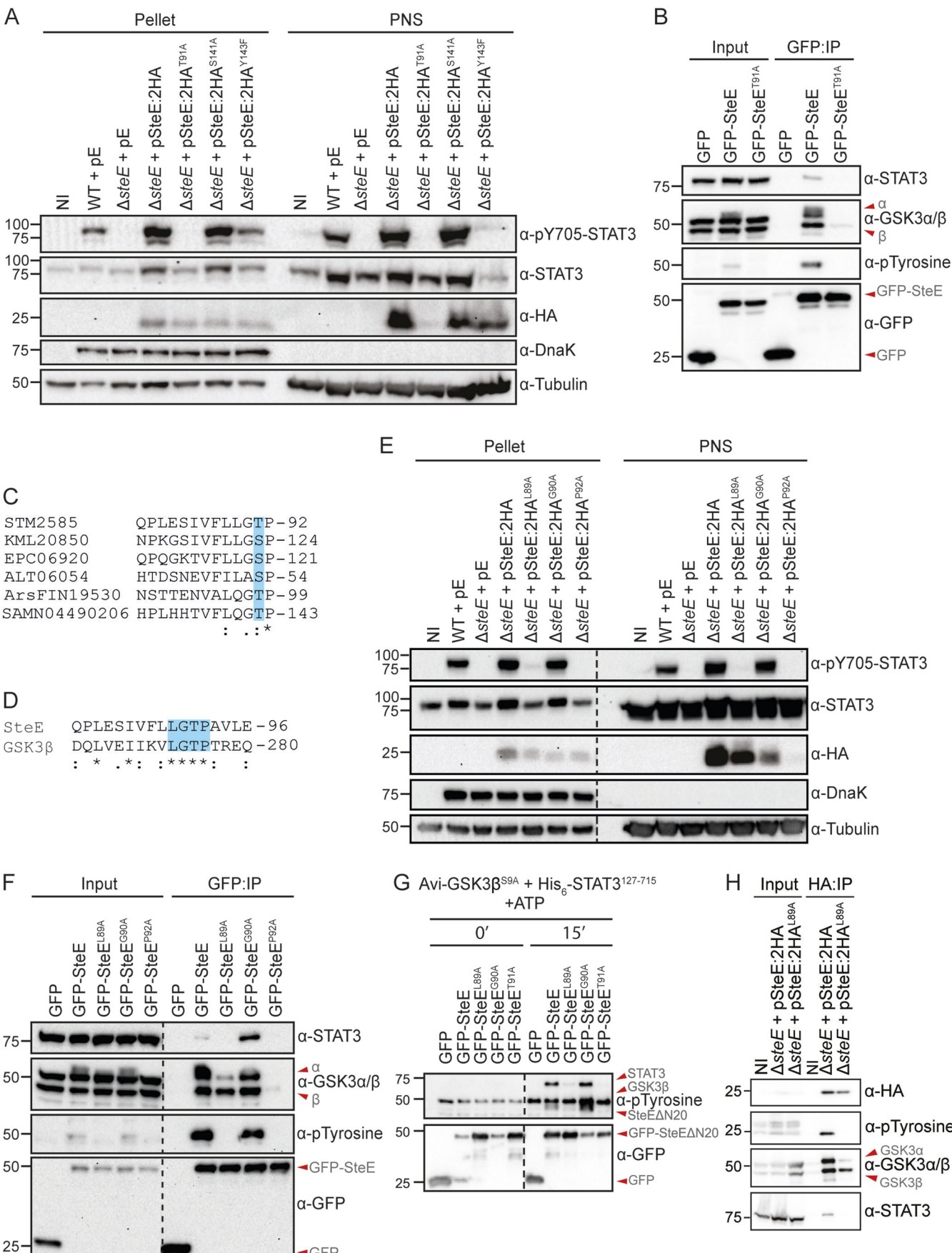

◄  **Figure 3.  SteE phosphorylation within a GSK3-like motif controls activity.**

(A) Cells infected with the indicated *Salmonella* strains for 17 h were lysed, with pellets and post-nuclear supernatants (PNS) separated by centrifugation. Samples were analysed by immunoblotting. NI non-infected, pE empty vector. Immunoblot represents three independent biological repeats. (B) 293ET cells expressing GFP or the indicated GFP-tagged SteE variant were lysed, and post-nuclear supernatants were subjected to GFP-TRAP immunoprecipitation (GFP:IP). Samples were analysed by immunoblotting for interaction to endogenous GSK3 and STAT3 and the tyrosine phosphorylation status of each GFP-tagged protein. Data represent three independent biological repeats. (C) Sequence alignment between SteE and its putative homologues around the T91 residue. Conservation of SteE T91 as threonine or serine in putative homologues is shown with shading. (D) Sequence alignment between SteE and GSK3β for the annotated amino acids. The LGTP motif is shown with shading. Asterisk (*) = fully conserved residue, Colon (:) = amino acid residues with similar properties, Period (.) = amino acid with weakly similar properties. (E) Cells infected with the indicated *Salmonella* strains for 17 h were lysed, with pellets and post-nuclear supernatants (PNS) separated by centrifugation. Samples were analysed by immunoblotting. NI non-infected, pE empty vector. Data represent three independent biological repeats. (F) 293ET cells expressing GFP or the indicated GFP-tagged SteE variant were lysed, and post-nuclear supernatants were subjected to GFP-TRAP immunoprecipitation (GFP:IP). Samples were analysed by immunoblotting for interaction to endogenous GSK3 and STAT3 and the tyrosine phosphorylation status of GFP-tagged protein. Data represent three independent biological repeats. See also Fig. EV4A. (G) *GSK3α/β*$^{-/-}$ 293ET cells expressing GFP or the indicated GFP-SteEΔN20 variant were lysed, and post-nuclear supernatants were subjected to GFP-TRAP immunoprecipitation (IP). GFP-tagged proteins immobilised on beads were assessed for their ability to be tyrosine phosphorylated by GSK3β in an in vitro kinase reaction containing 1 mM ATP, recombinant Avi-GSK3β$^{S9A}$ (0.2 μM) and His$_6$-STAT3$^{127-715}$ (0.2 μM). Data represent two independent biological repeats. See also Fig. EV4B. (H) Cells infected with the indicated *Salmonella* strains for 17 h were lysed, and post-nuclear supernatants were subjected to HA-TRAP immunoprecipitation (HA:IP). Input and IP samples were analysed by immunoblotting. NI non-infected. Data represent three independent biological repeats. Dotted lines on immunoblots represent lanes removed from the analysis, with all samples shown from one continuous membrane. Source data are available online for this figure.

activity of KML20850, we generated KML20850$^{FS-SP}$ and SteE$^{SP-FS}$ and monitored interaction with GSK and STAT3 as well as STAT3 phosphorylation. As expected, WT SteE interacted with GSK3 and STAT3 to form a complex and mutation of SP to FS reduced this interaction (Fig. 2D). However, more strikingly, whereas WT KML20850 showed very limited interaction with GSK3, mutation of FS to SP in KML20850 (KML20850$^{FS-SP}$) restored interaction with GSK3 and STAT3 phosphorylation despite no stable interaction with STAT3 (Fig. 2D,E). We propose that through molecular mimicry of a canonical GSK3 phosphorylation motif, the SLSLPVSP motif represents a newly defined SLiM that helps target this family of bacterial proteins towards the host serine/threonine kinase GSK3. That SteE$^{SP-FS}$ shows only reduced interaction with GSK3, whereas KML20850$^{FS-SP}$ restores the ability of KML20850 to interact with GSK3, raises the intriguing possibility that secondary low affinity interaction sites for GSK3 might be present in SteE. Whether these reside in the other protected regions identified by the HDX-MS analysis, awaits discovery.

## Successive SteE phosphorylation controls activity

As previously reported, SteE is a substrate of GSK3 (Panagi et al, 2020), with GSK3-dependent phosphorylation occurring at residues T91, S141 and Y143 of SteE. Whereas SteE phosphorylation at Y143 creates an SH2–binding motif (pYxxQ) that is recognised and bound by STAT3 (Gibbs et al, 2020) the importance of preceeding threonine and serine phosphorylation is unknown. Analysis of the SteE variants revealed consistently reduced threonine phosphorylation and absent tyrosine phosphorylation of both SteE$^{SLSL-AAAA}$ and SteE$^{SLPV-AAAA}$ when compared to WT SteE (Fig. 2F). This, together with the fact that GSK3 is canonically a serine/threonine kinase led us to hypothesise that (1) a stable interaction between SteE and GSK3 promotes GSK3-mediated tyrosine activity and (2) threonine phosphorylation of SteE is a prerequisite for tyrosine phosphorylation.

To probe the importance of GSK3-mediated threonine phosphorylation of SteE, we first analysed whether phospho-deficient SteE variants induced STAT3 phosphorylation following their translocation from bacteria. Immunoblot analysis revealed that WT SteE:2HA, SteE:2HA$^{S141A}$ and SteE:2HA$^{Y143F}$ were detected in the

post-nuclear fraction (cytosol) of infected cells and SteE:2HA$^{S141A}$ induced an equivalent level of STAT3 phosphorylation to WT SteE:2HA (Fig. 3A). As expected, cells infected with the strain carrying SteE:2HA$^{Y143F}$ showed almost negligible Y705-STAT3 phosphorylation in the post-nuclear supernatant fraction with reduced signal in the pellet fraction (Fig. 3A). This is in line with the function of phosphorylated Y143 in recruiting STAT3 to the SteE-GSK3 complex for phosphorylation (Gibbs et al, 2020). In contrast to WT SteE:2HA, the phospho-deficient variant SteE:2HA$^{T91A}$ was not detected in the cytosolic fraction of infected cells, despite a comparable expression to SteE:2HA within *Salmonella* (Fig. 3A, pellet fraction). Given our previous observation that WT SteE:2HA is unstable in *GSK3α/β*$^{KO}$ 293ET cells, after being translocated from *Salmonella* (Panagi et al, 2020), we hypothesised that mutation of T91 might prevent a stable interaction forming between SteE and GSK3. GFP-tagged SteE$^{T91A}$ remained stable upon its exogenous expression but unlike WT GFP-SteE, which formed a stable and specific complex with GSK3α/β and STAT3, the T91A substitution in GFP-SteE$^{T91A}$ disrupted its co-purification with both targets (Fig. 3B). Furthermore, immunoblot analysis revealed that no tyrosine phosphorylation of GFP-SteE$^{T91A}$ was detected (Fig. 3B). Together, our data reveal an essential requirement for residue T91 of SteE in both the formation of a stable SteE:GSK3 complex and ability of GSK3 to mediate tyrosine phosphorylation of SteE and STAT3, whereas S141 is not required for STAT3 phosphorylation.

## SteE mimics a eukaryotic motif to reprogramme the kinase activity of GSK3

GSK3 is a pleiotropic kinase with as many as 100 reported substrates, with all known canonical substrates phosphorylated on serine or threonine residues. Like other members of the CMGC kinase family, GSK3 undergoes an auto-tyrosine phosphorylation event on the activation-loop tyrosine during its maturation process (Lochhead et al, 2006). This one-off tyrosine phosphorylation event during translation of the CMGC kinases (Cole et al, 2004) is dependent on proline hydroxylation of the highly conserved L/xGxP motif (Lee et al, 2020). Intriguingly, SteE residue T91, which is essential for function, is conserved as either a threonine or serine in the homologues of SteE

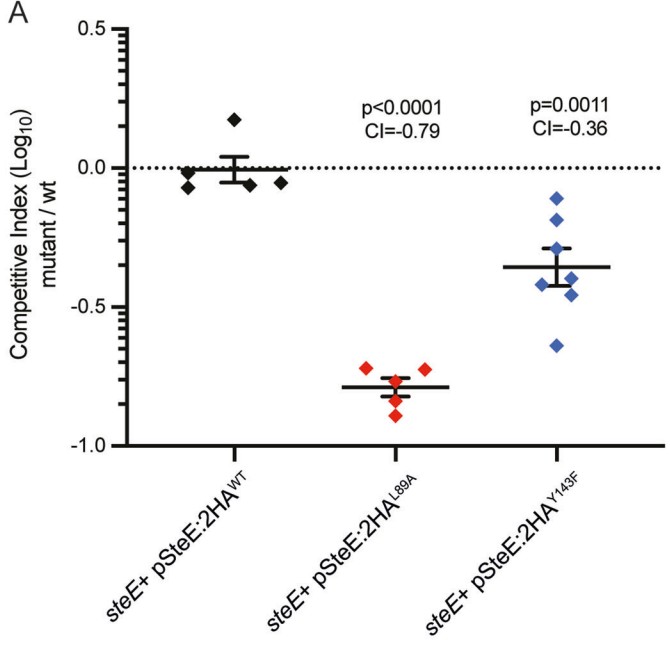

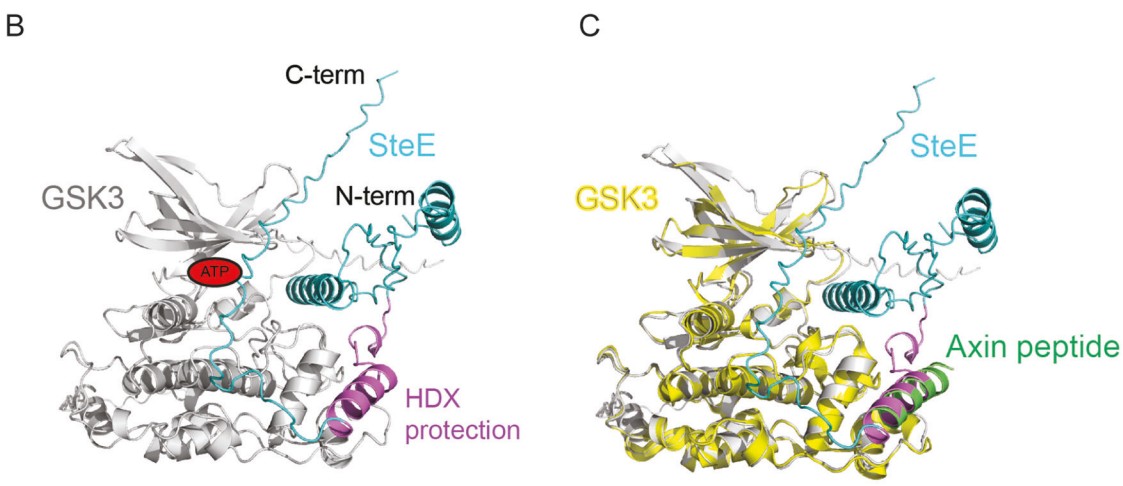

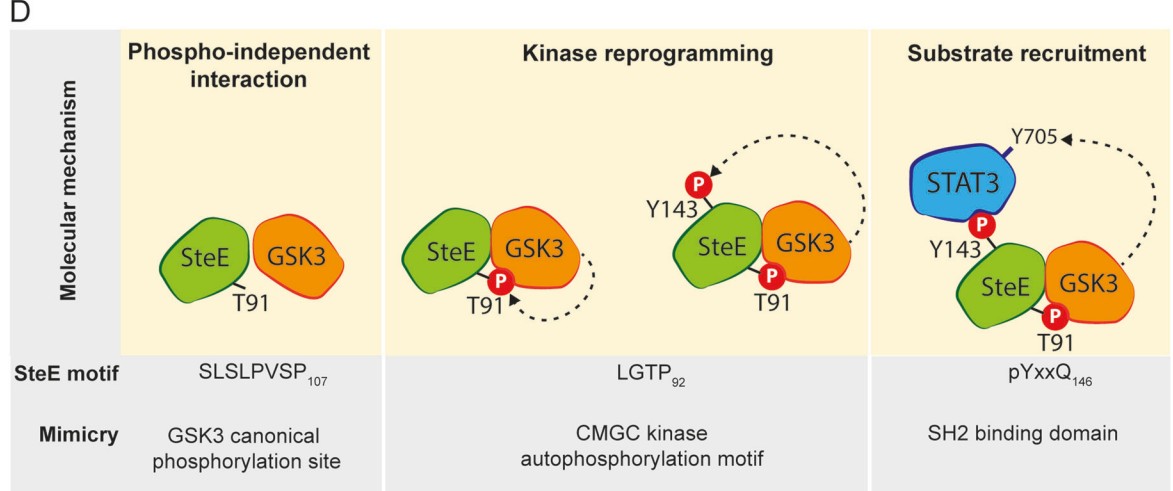

 **Figure 4. Mimicry of a GSK3-motif mediates kinase reprogramming and virulence.**

(A) C57BL/6 mice were inoculated by intraperitoneal injection with 500 CFU of each indicated bacterial strain. Bacteria were recovered from infected spleens 72 h post-inoculation and CI values calculated. The $\log_{10}$ CI is plotted with significance computed by one-way ANOVA and Dunnett's correction for multiple comparisons. Individual data points represent individual animals with $n = 5$ for WT and SteE$^{L89A}$ groups and $n = 7$ for the SteE$^{Y143F}$ group. Error bars denote the s.e.m. Comparisons were made to the wild-type SteE group with significance of $P < 0.0001$ for SteE$^{L89A}$ and $P = 0.0011$ for SteE$^{Y143F}$. (B) AlphaFold3 model of SteE (cyan) in complex with GSK3β (grey). Relative hydrogen/deuterium-exchange differences, i.e. protein regions experiencing protection upon complex formation, are highlighted in pink. The position of the ATP-binding pocket is indicated. (C) Overlap of the Alphafold3 prediction of the GSK3β/SteE complex with the crystal structure of the GSK3β/Axin-peptide complex (1o9u.pdb; yellow/green). (D) Proposed model of SteE-mediated reprogramming of GSK3. Source data are available online for this figure.

(Fig. 3C) and is embedded in an LGTP motif (Fig. 3D). To test whether the SteE-LGTP motif might revert/induce the tyrosine phosphorylation capability of GSK3 we mutated either L89, G90 or P92 of the LGTP motif to alanine and analysed the ability of these HA-tagged variants to induce STAT3 phosphorylation following infection of 293ET cells. Even though there was a slight reduction in the amount of translocated SteE$^{G90A}$, this SteE variant induced a similar degree of STAT3 Y705 phosphorylation when compared to WT SteE:2HA, suggesting it is not essential for the reprogramming of GSK3 kinase activity (Fig. 3E). In contrast, SteE$^{L89A}$ was unable to induce STAT3 phosphorylation, despite its stable translocation into the host cell cytosol. Interestingly, SteE$^{P92A}$ phenocopied SteE$^{T91A}$ in that this protein was unstable upon translocation from bacteria and therefore no STAT3 phosphorylation was detected (Fig. 3E).

To explore why SteE$^{L89A}$ was unable to induce STAT3 phosphorylation and to test the hypothesis that SteE$^{P92A}$ was unstable following its translocation from bacteria due to a lack of GSK3 interaction, we exogenously expressed GFP-tagged WT or mutant forms of SteE in 293ET cells and conducted a co-immunoprecipitation from cell lysates. Like WT GFP-SteE, GFP-SteE$^{G90A}$ formed a stable complex with both GSK3 and STAT3, which correlated with its ability to induce STAT3 phosphorylation (Fig. 3E,F). Interestingly, SteE$^{L89A}$ underwent threonine phosphorylation (Fig. EV4A) and retained the ability to interact with GSK3 but did not form a stable interaction with STAT3. The lack of stable interaction with STAT3 is explained by the absence of any detected tyrosine phosphorylation of GFP-SteE$^{L89A}$, which is in stark contrast to the phosphorylation status of WT SteE (Fig. 3F). Like SteE$^{T91A}$, exogenously expressed, GFP-tagged SteE$^{P92A}$ was expressed but unable to make a stable interaction with GSK3 (Fig. 3F).

To further explore the role of the LGTP motif in endowing GSK3 with the capability to phosphorylate substrates on tyrosine residues, each mutant form of SteE was tested in an in vitro kinase assay. GFP-tagged WT SteE$^{\Delta N20}$ and the indicated mutant forms of SteE$^{\Delta N20}$ were expressed in GSK3$\alpha/\beta^{KO}$ 293ET cells and immobilised by GFP-TRAP beads. Following incubation with recombinant Avi-GSK3β$^{S9A}$ and His-STAT3$^{127-715}$ for the indicated amount of time the phosphorylation status of SteE, GSK3 and STAT3 was determined by immunoblotting with an antibody against pTyrosine. WT and SteE$^{G90A}$ induced a similar amount of SteE and STAT3 tyrosine phosphorylation whereas SteE$^{L89A}$ and SteE$^{T91A}$ were strongly attenuated in their ability to induce the tyrosine phosphorylation capability of GSK3 towards either SteE or STAT3 (Figs. 3G and EV4B). Finally, the tyrosine phosphorylation status of SteE was monitored during infection. As expected, we detected tyrosine phosphorylation of WT SteE. In contrast, there was a complete loss of pTyrosine signal observed for SteE$^{L89A}$ and no interaction with STAT3 was detected (Fig. 3H). As with the analysis of transfected

SteE$^{L89A}$, translocated HA-tagged SteE$^{L89A}$ interacted with both isoforms of GSK3. Compared to WT SteE, there was a reduction in the stable interaction of SteE$^{L89A}$ with GSK3α (upper band in the the GSK3 blot) but not GSK3β. As SteE can function via GSK3α or GSK3β (Panagi et al, 2020) this observation is unlikely to explain the complete loss of tyrosine phosphorylation observed for SteE$^{L89A}$. In sum, we propose that the LGTP motif endows SteE with the capacity to interact with and reprogramme the kinase activity of GSK3 so that it phosphorylates tyrosine residues of the neosubstrates SteE and STAT3. We speculate that SteE, which shows a high degree of intrinsic disorder, uses a series of SLiMs to exploit a possible inherent plasticity with regards to GSK3's pleiotropic substrate repertoire and ability to mediate auto-tyrosine phosphorylation. Whether the LGTP motif of SteE might enable GSK3 to re-adopt the "prone-to-autophosphorylate" state through protein conformational changes requires further investigation that should include an analysis of phosphorylated SteE intermediates.

## Kinase reprogramming is essential for the virulence phenotype of SteE

SteE is required for *Salmonella* virulence and persistence (Pham et al, 2020) and *steE* mutant *S.* Typhimurium display a competitive disadvantage over wild-type bacteria in the mouse model of systemic disease (Gibbs et al, 2020). Even though tyrosine phosphorylation of SteE is required for virulence (Gibbs et al, 2020) it remains to be determined how a complete lack of kinase reprogramming impacts virulence. Therefore, the virulence of *steE* mutant bacteria expressing either WT SteE, SteE$^{Y143F}$ or SteE$^{L89A}$ against WT *S.* Typhimurium carrying an empty plasmid (pE) was analysed. Following intraperitoneal infection of mice with a 1:1 inoculum, *steE* mutant bacteria expressing WT SteE were as virulent as WT *S.* Typhimurium. In contrast, *steE* mutant bacteria expressing either SteE$^{L89A}$ or SteE$^{Y143F}$ were significantly attenuated in virulence compared to WT bacteria (Fig. 4A). These results demonstrate that SteE modulates *S.* Typhimurium virulence during systemic infection and that this is dependent on the mimicry of an eukaryotic LGTP motif that enables GSK3 to not only phosphorylate substrates on serine or threonine but also now on tyrosine residues.

## A series of conserved motifs regulate kinase reprogramming by SteE

In the absence of an experimental structure for the SteE-GSK3 complex we used AlphaFold3 to predict a model of the complex to aid interpretation of our data. The model with the highest interface predicted template modelling score (ipTM of 0.74) is shown in

Fig. 4B. This model predicts an extensive interface between the two proteins with SteE forming two α-helices that are part of the protein interface with additional contributions to the interface from unstructured regions of SteE. However, the predicted position error for most of the interface is high, with only the region covering amino acids 102-119 predicted with confidence (Fig. EV5A,B), which precludes a detailed analysis of the complex model. Encouragingly though, this predicted interface is supported by the HDX-MS and mutagenesis data presented here. The SLPVSP motif is located in an unstructured region that is predicted to lie at the interface with GSKβ and precedes α-helix (aa 107–119). Interestingly, this helix adopts a position similar to that of an axin-derived peptide in the crystal structure of a GSK3β-axin complex and, similar to the GSK3β-axin interaction, forms a hydrophobic interface (1o9u.pdb) (Figs. 4C and EV5C) (Dajani et al, 2003). Unfortunately, we did not have any peptide coverage for the region containing the LGTP motif in our HDX-MS experiments (Fig. EV2), which combined with the low confidence prediction score for this region prevents a structural interpretation of our mutagenesis data.

Collectively, using SteE as a prototype kinase reprogramming protein, we describe the molecular dissection of kinase reprogramming, a phenomenon that is controlled by a stepwise phosphorylation process on SteE and at least two newly defined motifs in this modulatory protein (Fig. 4D). The SLSLPVSP$_{107}$ motif mimics the canonical phosphorylation sequence of GSK3 to mediate interaction of "inert" unphosphorylated SteE with GSK3. Next, SteE becomes phosphorylated on T91, a conserved residue found within the newly identified 'tyrosine-kinase unlocking' LGTP motif in SteE. This LGTP motif, which resembles a motif found in several CMGC kinases that undergo auto-tyrosine phosphorylation, is essential for reprogramming the activity of GSK3, whereby SteE becomes phosphorylated on tyrosine residue(s) by GSK3. Finally, phosphorylation of residue Y143 of SteE generates an already described pYxxQ motif which interacts with the SH2 domain of STAT3 (Gibbs et al, 2020). Using biolayer interferometry, we confirmed that an SteE$^{138-148}$ peptide phosphorylated at Y143 interacted with recombinant His$_6$STAT3$^{127-715}$ and demonstrated that binding required Y143 phosphorylation but not S141 phosphorylation (Fig. EV5D).

Together, our data suggest that the combinatorial molecular mimicry of eukaryotic SLiMs is key to inducing the *trans* tyrosyl-directed phosphorylation activity of GSK3 (Fig. 4D). In general, SLiMs represent a form of kinase regulation that provides a means to expand the functionality of the conserved kinase core (Gógl et al, 2019). Combining this with phosphorylation-based switches in protein function enables the evolution of new kinase regulators, such as the family of kinase reprogramming proteins in this study. It is then also of interest that other pathogens have evolved phosphorylation-dependent mechanisms to tap into STAT3-mediated signalling pathways during infection (Panagi and Thurston, 2023).

Finally, that this process relies on the co-option of host motifs by SteE suggests the intriguing possibility that eukaryotes might use kinase reprogramming to regulate host cell signalling in other contexts. Together, our data challenge the dogma that GSK3 is a strict S/T-directed kinase and provide a mechanism by which bacterial effectors "restore" the dual specificity capacity of this kinase, further exploiting it for *trans*-phosphorylation of

neosubstrates. We propose that the directed targeting of intrinsically disordered proteins to kinases might provide an alternative means for phosphorylation-based therapeutics.

# Methods

## Reagents and tools table

| Reagent/resource | Reference or source | Identifier or catalogue number |
|---|---|---|
| **Experimental models** | | |
| 293ET cells | Gift from Felix Randow | RRID: CVCL_6996 |
| *GSK3α/β$^{-/-}$* 293ET cells | Panagi et al, 2020 | N/A |
| C57BL/6 mice | Envigo uk | N/A |
| *Salmonella enterica* serovar Typhimurium, wild-type (WT), strain 12023 | NCTC | NCTC:12023 |
| *steE* mutant (Δ*steE*) of *Salmonella enterica* serovar Typhimurium, strain 12023 | Panagi et al, 2020 | N/A |
| WT or Δ*steE Salmonella* (12023) carrying empty pWSK29 (pE) or pWSK29 for HA-tagged-SteE expression | Panagi et al, 2020 | N/A |
| Δ*steE* + p.SteE:2HA$^{L89A}$ | This study | N/A |
| Δ*steE* + p.SteE:2HA$^{G90A}$ | This study | N/A |
| Δ*steE* + p.SteE:2HA$^{T91A}$ | This study | N/A |
| Δ*steE* + p.SteE:2HA$^{P92A}$ | This study | N/A |
| Δ*steE* + p.SteE:2HA$^{S141A}$ | This study | N/A |
| Δ*steE* + p.SteE:2HA$^{Y143F}$ | This study | N/A |
| Δ*steE* + p.SteE:2HA$^{KESL-AAAA}$ | This study | N/A |
| Δ*steE* + p.SteE:2HA$^{SLSL-AAAA}$ | This study | N/A |
| Δ*steE* + p.SteE:2HA$^{SLSL-AAAA}$ | This study | N/A |
| Δ*steE* + p.SteE:2HA$^{SLPV-AAAA}$ | This study | N/A |
| Δ*steE* + p.SteE:2HA$^{SLPV-AAAA}$ | This study | N/A |
| Δ*steE* + p.SteE:2HA$^{QKL-AAA}$ | This study | N/A |
| *E. coli* DH5a | ThermoFisher | Cat#18265017 |
| *E. coli* TOP10 | ThermoFisher | N/A |
| *E. coli Rosetta (DE3)* | Millipore | Cat#70954 |
| *E. coli* BL21 (DE3) | New England | Cat#C2527 |
| SF9 insect cells | Invitrogen | Cat#10503433 |
| **Recombinant DNA** | | |
| ptCMV-GFP (pEGFP-N1) | Panagi et al, 2020 | N/A |
| ptCMV.GFP-SteE | Panagi et al, 2020 | N/A |
| ptCMV.GFP-SteE$^{ΔN20}$ | Panagi et al, 2020 | N/A |
| ptCMV.GFP-SteE$^{ΔN20 L89A}$ | This study | N/A |
| ptCMV.GFP-SteE$^{ΔN20 G90A}$ | This study | N/A |
| ptCMV.GFP-SteE$^{ΔN20 T91A}$ | This study | N/A |
| ptCMV.GFP-SteE$^{L89A}$ | This study | N/A |
| ptCMV.GFP-SteE$^{G90A}$ | This study | N/A |
| ptCMV.GFP-SteE$^{T91A}$ | Panagi et al, 2020 | N/A |
| ptCMV.GFP-SteE$^{P92A}$ | This study | N/A |
| ptCMV.GFP-SteE$^{S141A}$ | Panagi et al, 2020 | N/A |
| ptCMV.GFP-SteE$^{Y143F}$ | Panagi et al, 2020 | N/A |
| ptCMV.GFP-SteE$^{T91A/S141A}$ | This study | N/A |

| Reagent/resource | Reference or source | Identifier or catalogue number |
|---|---|---|
| ptCMV.GFP-SteE$^{T91A/Y143F}$ | This study | N/A |
| ptCMV.GFP-SteE$^{S141A/Y143F}$ | This study | N/A |
| ptCMV.GFP-SteE$^{T91A/S141A/Y143F}$ | Panagi et al, 2020 | N/A |
| ptCMV.GFP-SteE$^{KESL-AAAA}$ | This study | N/A |
| ptCMV.GFP-SteE$^{SLSL-AAAA}$ | This study | N/A |
| ptCMV.GFP-SteE$^{SLPV-AAAA}$ | This study | N/A |
| ptCMV.GFP-SteE$^{SP-FS}$ | This study | N/A |
| ptCMV.GFP-SteE$^{QKL-AAA}$ | This study | N/A |
| ptCMV.GFP-Epc06920 | This study | N/A |
| ptCMV.GFP-KML20850 | This study | N/A |
| ptCMV.GFP-KML20850$^{FS-SP}$ | This study | N/A |
| ptCMV.GFP-ALT06054 | This study | N/A |
| ptCMV.GFP-SAMN04490206 | This study | N/A |
| ptCMV.GFP-ArsFin19530 | This study | N/A |
| pET49-His$_6$-3C-SteE$^{\Delta N20}$ | This study | N/A |
| pET49-His$_6$-3C-STAT3$^{127-715}$ | This study | N/A |
| pACEBAC1-His$_6$-3C-GSK3β | This study | N/A |
| pACEBAC1-His$_6$-Avi-3C-GSK3β$^{S9A}$ | This study | N/A |
| **Antibodies** | | |
| Rabbit anti-pY705-STAT3 | Cell Signaling | Cat# 9145 |
| Rabbit anti-STAT3 | Cell Signaling | Cat#12640 |
| Rabbit anti-GSK3α/β | Cell Signaling | Cat#5676S |
| Mouse anti-pGSK3beta (Ser9) (Alexa 790-conjugated) | Santa Cruz Biotechnology | Cat#sc-373800 |
| Mouse anti-pGSK3 (Tyr216/279) | Sigma-Aldrich | Cat#05-413 |
| Rat anti-GFP | ChromoTek | Cat#3h9-100 |
| Mouse anti HA.11 | Bio Legend | Cat#901503 |
| Rabbit anti-His (HRP-conjugated) | Abcam | Cat#ab1187 |
| Mouse anti-Tubulin E7 | DSHB | Cat#E7 |
| Mouse anti-Tubulin | Gift from Prof Steve Ley, UCL | N/A |
| Mouse anti-DnaK | Enzo Life Sciences | Cat# ADI-SPA-880-D |
| Mouse anti-p-Tyrosine | Millipore | Cat# 05-321 |
| Mouse anti-p-Threonine | Cell Signaling | Cat# 9386 |
| Goat anti-Rabbit, HRP-linked | Agilent (Dako) | Cat#P0448 |
| Goat anti-Mouse, HRP-linked | Agilent (Dako) | Cat#P0447 |
| Goat anti-Rat, HRP-linked | Cell Signaling | Cat# 7077 |
| **Oligonucleotides and other sequence-based reagents** | | |
| PCR Primers | This study | Appendix Table S1 |
| Synthesised Genes | This study | Appendix Table S2 |
| **Chemicals, enzymes and other reagents** | | |
| ATP | ThermoFisher | Cat#R0441 |
| Lipofectamine 2000 | Life Technologies | Cat#11668019 |
| Cell Culture Lysis Buffer | Promega | #E153A |
| Kinase Buffer (10X) | Cell Signaling | 9802S |
| Immobilon®-P PVDF Membrane | Millipore | IPVH00010 |

| Reagent/resource | Reference or source | Identifier or catalogue number |
|---|---|---|
| ECL | GE Healthcare | RPN2209 |
| Pierce™ ECL Plus Western Blotting Substrate | ThermoFisher | #32132 |
| cOmplete™, Mini, EDTA-free Protease Inhibitor Cocktail | Roche | Cat#4693159001 |
| PhosSTOP™ | Roche | Cat#4906837001 |
| GFP-TRAP beads | ChromoTek | Cat#gta-100 |
| HA-TRAP beads | ChromoTek | Cat#ata-20 |
| HisPur Cobalt Superflow resin | ThermoFisherScientific | Cat#25229 |
| HisPur Ni-NTA resin | ThermoFisherScientific | Cat#88222 |
| Source 15S cation exchange chromatography resin | Cytiva Life Sciences | Cat#17094401 |
| HiTrap Q HP anion exchange chromatography column | Cytiva Life Sciences | Cat#17115401 |
| HiTrap Heparin HP chromatography column | Cytiva Life Sciences | Cat# 17040703 |
| Recombinant Avi-GSK3β$^{S9A}$ | This study | N/A |
| Recombinant GSK3β | This study | N/A |
| Recombinant His$_6$-SteE$^{\Delta N20}$ | This study | N/A |
| Recombinant His$_6$-STAT3$^{127-715}$ | This study | N/A |
| SteE_pY143 peptide, biotinylated | This study | The Francis Crick Institute |
| SteE_pS141 peptide, biotinylated | This study | The Francis Crick Institute |
| SteE_pS141pY143 peptide, biotinylated | This study | The Francis Crick Institute |
| **Software** | | |
| Prism Versions 9, 10 | GraphPad | https://www.graphpad.com/scientific-software/prism/ |
| Adobe Illustrator 2023, 2024 | Adobe Creative Cloud | https://www.adobe.com/uk/creativecloud.html |
| Image Lab Version 6.0.1 | Bio-Rad Laboratories | https://www.bio-rad.com/en-uk/product/image-lab-software?ID=KRE6P5E8Z |
| iBright Analysis Software Version 5.3.0 | ThermoFisher | https://www.thermofisher.com/uk/en/home/life-science/protein-biology/protein-assays-analysis/western-blotting/detect-proteins-western-blot/western-blot-imaging-analysis/ibright-systems/software.html |
| Pymol Version 2.3.1 | Schrödinger, LLC | www.pymol.org |
| TopSpin Version 4.1.4 | Bruker | www.bruker.com |
| CCPNmr Version 3.1.0 | CCPN | https://ccpn.ac.uk |
| J-815 Spectra Manager Version 2 | Jasco Corporation | |
| DichroWeb server for analysis of protein Circular Dichroism spectra | Ref 30 | http://dichroweb.cryst.bbk.ac.uk/html/home.shtml |
| DynamX HDX data analysis software Version 3.0 | Waters | www.waters.com |
| Octet BLI Discovery Versions 12.2.2.20 and 12.2.2.24 | Sartorius | https://www.sartorius.com/en |
| Clustal Omega (1.2.4) | Ref 26 | https://www.ebi.ac.uk/jdispatcher/msa/clustalo |

| Reagent/resource | Reference or source | Identifier or catalogue number |
|---|---|---|
| AlphaFold2 | Google DeepMind/ The Francis Crick Institute local installation | https://github.com/ google-deepmind/ alphafold |
| **Other** | | |
| GenElute™ Plasmid Miniprep Kit | Sigma | PLN350 |
| QIAquick PCR Purification Kit | Qiagen | #28106 |
| QIAquick Gel Extraction Kit | Qiagen | #28706 |
| ChemiDoc Imaging System | Bio-Rad Laboratories | https://www.bio-rad.com/ en-uk/category/chemidoc- imaging-systems? ID=NINJ0Z15 |
| iBright Imaging System | ThermoFisher | https:// www.thermofisher.com/ uk/en/home/life-science/ protein-biology/protein- assays-analysis/western- blotting/detect-proteins- western-blot/western-blot- imaging-analysis/ibright- systems.html |

## Cell culture

HEK293ET (originating from a female foetus), were provided by Dr Felix Randow and were maintained in Dulbecco's modified Eagle's medium (DMEM; Sigma) supplemented with 10% foetal calf serum (FCS; GIBCO, Life Technologies) at 37 °C in 5% $CO_2$. Cells were transfected using Lipofectamine 2000 (Life Technologies, Inc.) as per the manufacturer's instructions. Cells were routinely tested for Mycoplasma.

## Bacterial strains

*Escherichia coli* strain DH5α was used for cloning (ThermoFisher), and BL21 (DE3) (New England) for protein expression. *Salmonella enterica* serovar Typhimurium strain NCTC 12023 was used. The *steE* mutant was previously described (Panagi et al, 2020). Bacteria were grown overnight in LB at 37 °C with shaking.

## Mouse ethics statement

Animal experiments were performed in accordance with ASPA and UK Home Office regulations, and animals were given at least one week to acclimatise to the on on-site Animal Research Facility. The project licence for animal research (P2ED1F62A) was approved by Imperial College London Animal Welfare and Ethical Review Body (ICL AWERB).

## DNA plasmids and cloning

Primers for genes and associated truncations or point mutations generated in this study are provided in Appendix Table S1. For the transient expression of GFP-tagged proteins in mammalian cells, pTCMV-GFP-[gene] plasmids were used (Panagi et al, 2020). pWSK29 was used for the expression of HA-tagged effectors in *Salmonella* strains. Mammalian (ptCMV) and bacterial (pWSK29) expression vectors were constructed by restriction enzyme

digestion, followed by T4 DNA ligase-mediated ligation (NEB) of the cleaved insert and the linearised plasmid backbone. The gene of interest (insert) was generated by PCR-amplification of a suitable template—plasmid, *S.* Typhimurium genomic DNA, or synthesised gene (for SteE homologues, Appendix Table S2) - using primers containing restriction sites compatible for cloning in ptCMV-GFP or pWSK29. Following restriction enzyme-mediated digest, the insert was ligated in a ptCMV-GFP vector linearised using PciI and NotI or a pWSK29 vector linearised with EcoRI and BamHI. In ptCMV-GFP, the GFP coding sequence is upstream of PciI and NotI restriction sites, enabling the addition of the gene of interest downstream of GFP to generate a GFP-insert fusion. In pWSK29-[gene]-2HA, the EcoRI and BamHI restriction sites are found between the *ssaG* (SPI-2) promoter and the coding sequence of a double haemagglutinin (2HA) tag. All generated plasmids were transformed in *E. Coli* DH5α cells by the heat-shock method. Briefly, the ligation reaction mixture and 50 μl of bacterial suspension were gently mixed and incubated on ice for 2–5 min and then transferred at 42 °C for 45 s. Heat-shocked bacteria were immediately recovered in 500 μl SOC medium (0.5% Yeast extract, 2% Tryptone, 10 mM NaCl, 2.5 mM KCl, 10 mM $MgCl_2$, 10 mM $MgSO_4$ and 20 mM Glucose) and grown for 1–2 h at 37 °C with shaking. After centrifugation at $12,000 \times g$ for 1 min, bacterial pellets were resuspended in ~50 μl SOC medium, spread on LB agar plates supplemented with 100 μg/mL of the appropriate antibiotic and grown overnight at 37 °C. Single bacterial colonies were grown overnight in liquid LB and DNA plasmid was extracted using the GenElute™ Plasmid Miniprep Kit (Sigma), following the manufacturer's instructions. The purified pWSK29 plasmid was subsequently transformed in S. Typhimurium (refer to *Salmonella* transformation).

Recombinant *Salmonella* Typhimurium His$_6$-SteE$^{\Delta N20}$ and human His$_6$-STAT3$^{127-715}$ were ligated into pET49 vector using Gibson Assembly; the N-terminal His$_6$-tag is cleavable with HRV 3 C protease.

Human GSK3β and GSK3β$^{S9A}$ were cloned with an HRV 3C protease-cleavable N-terminal His$_6$-tag into pACEBac1 vector using Gibson Assembly. To make the His$_6$-avi-3C-GSK3β$^{S9A}$ construct, subsequently referred to as Avi-3C-GSK3β$^{S9A}$, an avi-tag was inserted as previously described (Fairhead and Howarth, 2015). Transposition and bacmid preparation were carried out as described in sections D.4. and D.5. of the MultiBac™ user manual (https://doi.org/10.13140/2.1.2563.6645). Briefly, 10 ng of the pACEBac1 vector containing the gene of interest was transposed into 100 μL *E. coli* DH10EMBacY cells and plated on agar supplemented with 50 μg/mL kanamycin, 7 μg/mL gentamycin, 10 μg/mL tetracyclin, 40 μg/mL IPTG and 100 μg/mL BluoGal. After 48 h of incubation at 37 °C white colonies were picked for multiplication overnight in 5 ml of LB media containing 50 μg/mL kanamycin, 7 μg/mL gentamycin and 10 μg/mL tetracyclin at 37 °C, 220 rpm shaking speed. Bacmid DNA was extracted using a combination of QIAprep® Spin Miniprep Kit for cell lysis and precipitation of DNA with isopropanol from cleared lysate.

All plasmids constructed in this study were sequence verified (GATC Biotech and Azenta/Genewiz).

## *Salmonella* transformation

*S.* Typhimurium strains grown to stationary phase overnight were sub-cultured at 1:100 dilution in 20 mL fresh LB supplemented with 50 μg/mL of the appropriate antibiotic and grown to OD 0.4–0.6 at

37 °C with shaking. Bacterial sub-cultures were cooled on ice for 15 min and pelleted by centrifugation at $2000 \times g$ at 4 °C for 10 min. Bacterial pellets were washed three times in sterile, ice-cold water and once in sterile, ice-cold 10% glycerol, before resuspension in 250 μL 10% glycerol. For transformation, 75 μl of bacterial suspension was gently mixed with 100 ng purified pWSK29 plasmid and electroporated within a 2 mm electroporation cuvette at 2.5 V, 25 μF, 200 ohms (Gene Pulser 2, Bio-Rad). Electroporated bacteria were recovered in 500 μl SOC medium for 1–2 h at 37 °C with shaking. Thereafter, bacteria pellets were plated on LB agar plates, supplemented with 100 μg/mL of the appropriate antibiotics, and grown overnight at 37 °C.

## Preparation of viral stock for expression of recombinant avi-GSK3β$^{S9A}$ in Sf9 insect cells

Sf9 insect cells (Invitrogen Cat#10503433) were grown in Sf-900™ III SFM medium (Gibco™) at 28 °C and 150 rpm. For virus generation, the cells were split to $0.4 \times 10^6$ cells/mL in 2 mL in a six-well plate and left to settle for 30 min. Approximately 3 μg/μL bacmid DNA was transfected into Sf9 insect cells using Cellfectin II or Flyfectin reagent according to the manufacturer's instructions. Bacmid DNA contained a GFP gene to allow evaluation of transfection success. After 72 h incubation 500 μL of cell supernatant was used to infect 25 mL of Sf9 insect cells at ca. $3 \times 10^6$ cells/mL. Viral stock was obtained after 72 h incubation by removal of cells by centrifugation (700 rcf, 20 °C, 7 min). The last step was repeated up to two times but using only 50 μL of viral stock per infection.

## Protein expression and purification

Recombinant His$_6$-SteE$^{ΔN20}$ and His$_6$-STAT3$^{127-715}$ were expressed in BL21 (DE3) *E. coli* cells grown in LB medium supplemented with Kanamycin (50 μg/mL) at 37 °C. When the OD reached 0.8 protein expression was induced with 0.5 mM IPTG and cells were incubated at 18 °C for 16 h. Cultures were harvested by centrifugation at 4500 rcf and 4 °C for 20 min. Cell pellets were resuspended in 50 mM Tris, 300 mM NaCl, 20 mM imidazole, 0.5 mM TCEP, pH 8 at 4 °C (buffer A), added with 10 mM MgCl$_2$, 10% glycerol, 1 mM AEBSF, 1 PI tablet/50 mL buffer, 1.5 μg/mL DNase I, and lysed either by sonication (SteE) or 3 passes at 15,000 PSI over an Emulsiflex-05 homogeniser (STAT3). Lysate was cleared by ultracentrifugation ($53,343 \times g$, 4 °C, 1 h) and mixed with Ni-NTA resin pre-equilibrated with buffer A for immobilised metal ion affinity chromatography (IMAC). Following incubation for 30 min at 4 °C the lysate was drained and the resin washed with ca. 10 column volumes (CV) of 50 mM Tris, 1 M NaCl, 35 mM imidazole, 0.5 mM TCEP, pH 8 at 4 °C and ample buffer A. His$_6$-STAT3$^{127-715}$ was eluted with 50 mM Tris, 250 mM NaCl, 300 mM imidazole, 0.5 mM TCEP, 5% glycerol, pH 8 at 4 °C. His$_6$-SteE$^{ΔN20}$ was eluted with 50 mM Tris, 250 mM NaCl, 250 mM imidazole, 0.5 mM TCEP, pH 8 at 4 °C. To obtain untagged SteE$^{ΔN20}$ the His$_6$-tag was cleaved off with home-made HRV C3 protease (ca. 40 μg/50 mL) whilst dialysing the protein back into buffer A overnight at 4 °C for reverse IMAC to remove both tag and protease. Next, His$_6$-SteE$^{ΔN20}$/SteE$^{ΔN20}$ and His$_6$-STAT3$^{127-715}$ were subjected to anion exchange chromatography using 5 mL Q and Heparin columns, respectively. To ensure protein binding to these columns the NaCl

concentration was carefully reduced to 50 mM by fivefold dilution with salt-free buffer (25 mM Tris, 0.5 mM TCEP, 5% glycerol, pH 8 at 4 °C for His$_6$-STAT3$^{127-715}$; 25 mM Tris, 0.5 mM TCEP, 2% glycerol, pH 7.5 at 4 °C for His$_6$-SteE$^{ΔN20}$/ SteE$^{ΔN20}$). Proteins were syringe-filtered (0.22 μm) prior to loading at 2.5 mL/min flow. Unbound protein was washed off with 25 mM Tris, 50 mM NaCl, 0.5 mM TCEP, 5% glycerol at pH 8 at 4 °C (His$_6$-STAT3$^{127-715}$) or 25 mM Tris, 50 mM NaCl, 0.5 mM TCEP at pH 7.5 at 4 °C (His$_6$-SteE$^{ΔN20}$/SteE$^{ΔN20}$) and protein eluted by linear gradient from 50 mM to 1 M NaCl over 16 CV (His$_6$-STAT3$^{127-715}$) or 50–750 mM over 20 CV (His$_6$-SteE$^{ΔN20}$/SteE$^{ΔN20}$) at 3 mL/min flow. Purest fractions by SDS gel were pooled and further purified by size-exclusion chromatography into 25 mM HEPES, 200 mM NaCl, 10 mM MgCl$_2$, 0.5 mM TCEP, pH 8 at 4 °C (S200 16/600, His$_6$-STAT3$^{127-715}$) or 25 mM HEPES, 150 mM NaCl, 0.5 mM TCEP, pH 7.2 at 12 °C (S75 16/600, His$_6$-SteE$^{ΔN20}$/SteE$^{ΔN20}$). Proteins were concentrated (maximum 6 mg/mL for His$_6$-STAT3$^{127-715}$), flash-frozen in liquid nitrogen and stored at −80 °C.

Recombinant GSK3β and Avi-GSK3β$^{S9A}$ were expressed in SF9 insect cells (Invitrogen Cat#10503433) (infected with 1 mL viral stock/0.5 L culture and incubated at 28 °C and 150 rpm shaking speed. Cells were harvested 72 h h post infection (1200 rcf, 4 °C, 20 min) and pellets were frozen at −20 °C prior to purification. Purification of His$_6$-GSK3β and Avi-3C-GSK3β$^{S9A}$ is identical except for buffer pH during affinity chromatography, which was 7.8 at 4 °C for His$_6$-GSK3β buffers and 7.5 at 4 °C for Avi-3C-GSK3β$^{S9A}$. Cell pellets were resuspended in 50 mM HEPES, 300 mM NaCl, 0.5 mM TCEP (buffer A) with 10% glycerol, 10 mM MgCl$_2$, 1 complete mini EDTA-free protease inhibitor tablet/50 mL, 1 mM AEBSF, and 2 μg/mL DNase I, and lysed by three passes over an EmulsiFlex-05 homogeniser at 10,000–15,000 PSI. Lysate was cleared by ultracentrifugation ($53,343 \times g$, 4 °C, 1 h) and protein was purified by immobilised metal ion affinity chromatography (IMAC) using cobalt resin. The resin was incubated with lysate for ca. 30 min at 4 °C, drained and washed with 50 mM HEPES, 1 M NaCl, 0.5 mM TCEP and buffer A (combined ca. 100 CV). Proteins were eluted with 50 mM HEPES, 250 mM NaCl, 250 mM imidazole, 0.5 mM TCEP (buffer B). To remove the His$_6$-tag, HRV C3 protease was added to the His$_6$-GSK3β eluate (ca. 40 μg/50 mL) before dialysis into buffer A overnight at 4 °C and reverse IMAC. GSK3ß and Avi-3C-GSK3β$^{S9A}$ were further purified by cation exchange chromatography to separate different phospho-species. Reverse IMAC flowthrough (GSK3β) or IMAC eluate (Avi-3C-GSK3β$^{S9A}$) were diluted 1:5 with 25 mM HEPES, 0.5 mM TCEP, pH 7.0 at 4 °C to a final NaCl concentration of 50 mM, syringe-filtered (0.22 μm) and loaded onto a home-packed Source 15S cation exchange chromatography column (2.5 mL/min). Unbound protein was washed out with 3 CV of 25 mM HEPES, 50 mM NaCl, 0.5 mM TCEP, pH 7.0 at 4 °C before applying a linear NaCl gradient from 50 to 500 mM over 20 CV at 3 mL/min flow. The phosphorylation state of fractions containing GSK3β or Avi-GSK3β$^{S9A}$ was evaluated by dot blot analysis using anti-GSK3β, anti-pGSK3β (Tyr216) and anti-pGSK3β (Ser9) antibodies. Tyr216 monophosphorylated GSK3β or Avi-3C-GSK3β$^{S9A}$ was then buffer-exchanged into 25 mM HEPES, 200 mM NaCl, 0.5 mM TCEP, pH 7.2 at 12 °C by size-exclusion chromatography (S200 16/600). Fractions containing pure protein based on SDS gel were concentrated to ~6.5 mg/mL and flash-frozen in liquid nitrogen. Sample purity and phosphorylation state was confirmed by intact mass spectrometry.

## NMR spectroscopy

### Expression of isotopically labelled His_6-SteE^{ΔN20}/SteE^{ΔN20}

All additive solutions were sterile-filtered before addition to autoclaved minimal medium. $^{15}N$-isotopically labelled SteE^{ΔN20} was overexpressed in 2xM9 minimal medium with greater buffering capacity (Azatian et al, 2019) consisting of 13.6 g/L Na_2HPO_4, 6 g/L KH_2PO_4, 1 g/L NaCl supplemented with 2 mM MgSO_4, 1 mL/L of each trace element solution 1 and 2 (MP Cat#1676649), 50 µg/mL Kanamycin, 4 g/L glucose and 0.7 g/L $^{15}NH_4Cl$ (15N, 99%). Expression conditions were the same as for native SteE^{ΔN20} and purification differed as follows: the anion exchange chromatography was omitted and the size-exclusion chromatography (SEC) buffer contained 25 mM HEPES, 100 mM NaCl, 5 mM DTT, 2.5 mM MgCl_2 at pH 7.0 at 25 °C. Isotopically labelled His_6-SteE^{ΔN20} was purified as described for native SteE^{ΔN20} but without cleaving off the tag and buffer-exchanging it into 20 mM Tris-HCl, 100 mM NaCl, 1 mM DTT, 1 mM MgCl_2 at pH 7.0. The proteins were dialysed into their respective SEC buffers and 10% D_2O was added as lock solvent prior to data acquisition.

### Data acquisition

The 1D $^1H$ NMR spectrum was acquired on a Bruker Avance Neo 950 MHz spectrometer with 5 mm 1H/13C/15N cryoprobe, and 2D $^1H,^{15}N$-sofast HMQC spectra were recorded at 298 K at 250-266 µM concentration on a Bruker AVANCE III HD 700 MHz spectrometer equipped with a 5 mm 1H/31P/13C/15N quadruple-resonance PFG cryoprobe with cooled 13C/31 P preamplifiers. Spectra were processed with Bruker TopSpin software version 4.1.4 and figures were made with CCPNmr Analysis 3.1.0 and Adobe Illustrator 2023.

## Size-exclusion chromatography

Recombinant Y216-monophosphorylated GSK3β was mixed with SteE^{ΔN20} in a 1:1.1 molar ratio and added with 2 mM AMPPNP was added before loading the mixture on an S200 16/600 size-exclusion chromatography column. Gel filtration was carried out in 25 mM HEPES, 200 mM NaCl, 10 mM MgCl_2, 0.5 mM TCEP, pH 7.3 at 4 °C. Protein elution was monitored by UV absorbance at 280 nm and peak fractions were analysed by SDS gel electrophoresis.

## Biolayer interferometry analysis

Binding affinities of His_6-STAT3^{127-715} to SteE peptides were derived from association rate constants measured on an Octet R8 system (Sartorius) at 25 °C. Biotinylated SteE peptides were synthesised by the Chemical Biology Science Technology Platform at the Francis Crick Institute and dissolved from lyophilised stock in the respective protein buffer to which 0.005% Tween-20 was added (BLI buffer). Appropriate serial dilutions of His_6-STAT3^{127-715} were prepared in 25 mM HEPES, 200 mM NaCl, 10 mM MgCl_2, 0.5 mM TCEP at pH 7.7. SteE peptides were immobilised on Streptavidin (SA) sensors at 0.2 µg/mL by incubation until the linear signal increase started to plateau (up to 160 s), followed by a sensor wash in BLI buffer for 50 s to stabilise the baseline. Association was recorded by incubating the sensors with His_6-STAT3^{127-715} for 300 s. Subsequent dissociation in BLI buffer was followed for up to 300 s. Data were acquired using Octet BLI Discovery software 12.2.2.20

and processed with Octet BLI Analysis software 12.2.2.4 by subtraction of reference wells without protein and y alignment of the data to the baseline. Rate curves were calculated separately for each association curve (local fit) using a 1:1 model for steady-state analysis. The resulting concentration-dependent response values at equilibrium were exported to GraphPad Prism 10 and fitted using the equation for one-site specific binding to obtain $K_d$ values.

## CD spectroscopy

Far-UV circular dichroism spectroscopy of His_6-SteE^{ΔN20} was carried out on a Jasco J-815 CD spectropolarimeter (Jasco Inc.) with PMT detector using a Quartz cuvette with 0.1 cm pathlength (Hellma Analytics). Wavelength spectra for secondary structure analysis were recorded at 20 °C between 190 and 260 nm in 0.2 nm increments at a concentration of 0.15 mg/mL in 10 mM phosphate buffer (KPi) with 100 mM Na_2SO_4 and 0.5 mM TCEP at pH 7. A corresponding buffer spectrum was subtracted from the His_6SteE^{ΔN20} spectrum for baseline correction before ellipticities Θ were converted into mean residue ellipticity $[\Theta]_{MR}$ to normalise the values for the protein concentration:

$$\Theta_{MR} = \frac{\Theta * MRW}{10 * L * c} [\text{deg cm}^2 \text{dmol}^{-1}]$$

where Θ is the CD signal (in millidegrees), L is the pathlength (in cm), c is the concentration in mg/ml, and MRW is the mean residue weight (molecular weight divided by the number of residues). The processed wavelength spectrum was submitted to the DichroWeb Server (Miles et al, 2022; Whitmore and Wallace, 2004, 2008) for secondary structure analysis using reference dataset 7 (Sreerama and Woody, 2000; Sreerama et al, 2000) in combination with Contin-LL (Provencher and Glöckner, 1981; van Stokkum et al, 1990), CDSSTR (Compton and Johnson, 1986; Manavalan and Johnson, 1987), and Selcon3 (Sreerama and Woody, 1993; Sreerama et al, 1999) algorithms for deconvolution.

## Co-immunoprecipitations

293ET cells expressing GFP-tagged proteins were lysed in ice-cold Lumier's Lysis buffer (20 mM Tris-HCl pH 7.4, 150 mM NaCl, 10% glycerol, 0.3% Triton X-100) supplemented with a protease inhibitor cocktail (Roche buffer) and PhosSTOP (Roche). Cells were lysed on ice for 10–15 min, with periodic vortexing, before clarification by centrifugation at $17,000 \times g$ at 4 °C for 10 min. In all, 1/8th or 1/10th of the resulting supernatant was taken as the Input sample. The remaining supernatant was incubated with GFP-Trap beads (Chromotek) that were pre-washed three times in Lumier's Lysis Buffer, before equilibration in the same buffer. Post-nuclear supernatants were incubated with the beads for at least 2 h, at 4 °C and with rotation. Afterwards, beads were washed 4 times in 1 mL ice-cold Lumier's Lysis Buffer (as above but without protease and phosphatase inhibitors). Bound proteins were eluted from beads by the addition of 2× SDS loading buffer (125 mM Tris-Cl pH 6.8, 4% SDS, 10% glycerol, bromophenol blue (with 5% freshly added β-Mercaptoethanol]) and boiling at 95 °C for 7 min. Input samples were denatured in the same manner, and the resulting samples – Input (clarified lysate) and IP (bound) were analysed by SDS-PAGE and immunoblotting. The same protocol was followed

for HA based immunoprecipitation except that cells were infected with the indicated *Salmonella* strains as described and analysed at 17 h post infection.

## Competitive index

Mixed infections were carried out in female C57BL/6 mice (7–8 weeks old) with animals inoculated intraperitoneally with a 1:1 mixture of two strains comprising a total of 1000 colony-forming units delivered in PBS. The competitive index (CI) was determined from spleen homogenates 72 h post-inoculation as described previously (Beuzon et al, 2000) using a S. Typhimurium strain 12023 carrying an empty pWSK29 plasmid as the wild-type strain. Those harvesting spleens and extracting bacteria were blind to sample group. One-way ANOVA corrected by Dunnett's multiple comparison test was used to compare the $\log_{10}$ CI to the group analysing *steE* + pSteE:2HA$^{WT}$.

## Hydrogen/deuterium exchange

### Sample preparation and data acquisition
For HDX-MS experiments, recombinant GSK3β and SteE$^{\Delta N20}$ samples were purified into 25 mM HEPES, 200 mM NaCl, 0.5 mM TCEP, pH 7.2 to which 10 mM MgCl$_2$ and 0.5 mM AMPPNP were added. To obtain the complex proteins were mixed at final equimolar concentrations of 12 µM.

HDX-MS experiments were performed with final equimolar concentration of 5 µM. An aliquot of 5 µl was incubated with 45 µL of D$_2$O buffer at RT for 3, 30, 300 and 3000 s in triplicate. The labelling reaction was quenched by adding chilled 2.4% v/v formic acid in 2 M guanidinium hydrochloride and immediately frozen in liquid nitrogen. Samples were stored at −80 °C prior to analysis.

The quenched protein samples were rapidly thawed and subjected to proteolytic cleavage by pepsin followed by reversed phase HPLC separation. Briefly, the protein was passed through an Enzymate BEH immobilised pepsin column, 2.1 ×30 mm, 5 µm (Waters, UK) at 200 µL/min for 2 min and the peptic peptides trapped and desalted on a 2.1 ×5 mm C18 trap column (Acquity BEH C18 Van-guard pre-column, 1.7 µm, Waters, UK). Trapped peptides were subsequently eluted over 12 min using a 5-36% gradient of acetonitrile in 0.1% v/v formic acid at 40 µL/min. Peptides were separated on a reverse phase column (Acquity UPLC BEH C18 column 1.7 µm, 100 mm × 1 mm (Waters, UK). Peptides were detected on a SELECT SERIES Cyclic IMS mass spectrometer (Waters, UK), acquiring over a *m/z* of 300–2000, with the standard electrospray ionisation (ESI) source and lock mass calibration using [Glu1]-fibrino peptide B (50 fmol/µL). For protein identification, mass spectra were acquired in MS$^e$ mode. The mass spectrometer was operated at a source temperature of 80 °C and a spray voltage of 3.0 kV. Spectra were collected in positive ion mode. The other spectrometer parameters were as follows: cone voltage 30 V, desolvation gas 650 (L/h), collision energy ramp 20–40 V. HDX experiments were done in triplicate.

### Data analysis
Peptide identification was performed by MS$^e$ (Silva et al, 2005) using an identical gradient of increasing acetonitrile in 0.1% v/v formic acid over 12 min. The resulting MS$^e$ data were analysed

using Protein Lynx Global Server software (Waters, UK) with an MS tolerance of 5 ppm.

Mass analysis of the peptide centroids was performed using DynamX software (Waters, UK) with the following thresholds: minimum products per amino acid 0.3, minimum consecutive products 1, maximum ppm 5. Only peptides with a score >6.4 were considered. The first round of analysis and identification was performed automatically by the DynamX software, however, all peptides (deuterated and non-deuterated) were manually verified at every time point for the correct charge state, presence of overlapping peptides, and correct retention time. Deuterium incorporation was not corrected for back-exchange and represents relative, rather than absolute changes in deuterium levels. Changes in H/D amide exchange in any peptide may be due to a single amide or several amides within that peptide. All time points in this study were prepared at the same time and individual time points were acquired on the mass spectrometer on the same day. The values reflecting the experimental mass of each peptide in all possible states, replicates, time points, and charge states were exported from DynamX 3.0. H/D exchange differences were mapped onto Alphafold2 models of the GSK3β:SteE$^{\Delta N20}$ complex using Pymol. Figures were made with Adobe Illustrator.

## Alphafold multimer analysis of GSK3β-SteE complex

Protein structure prediction was performed using The Francis Crick Institute local installation of AlphaFold3 (https://github.com/deepmind/alphafold). The software was run with default parameters on a high-performance computing cluster. Input sequences were provided in FASTA format, and the predictions were generated using the full database mode.

## In vitro kinase assay

To monitor phosphorylation in vitro, *GSK3α/β$^{KO}$* 293ET cells (Panagi et al, 2020) seeded in a 6-well format were transfected with 800 ng ptCMV.GFP, ptCMV.GFP-SteE$^{\Delta N20}$ or the required ptCMV.GFP-SteE$^{\Delta N20}$ mutant, with four wells used per transfection condition. After 48 h, cells from each transfection condition were pooled together and lysed in 1.4 mL Lysis Buffer (150 mM NaCl, 0.3% Triton X-100, 20 mM Tris-Cl pH 7.4, 5% Glycerol, 5 mM EDTA), supplemented with a Protease Inhibitor Cocktail (cOmplete™, Roche), and clarified by centrifugation at $17,000 \times g$ for 10 min at 4 °C. GFP-tagged proteins from post-nuclear supernatants were then immobilised on beads (as described above). Beads were then washed three times in 1 mL Lysis Buffer and twice in 1 mL 1× Kinase Buffer (Cell Signaling) supplemented with Phosphatase Inhibitors (PhosSTOP, Roche). Beads containing GFP-tagged targets were then resuspended in Kinase Buffer containing 1 mM ATP (ThermoFisher) with or without 0.2 µM of non-phosphorylated Avi-GSK3β$^{S9A}$ and 0.2 µM His$_6$-STAT3$^{127-715}$ (Panagi et al, 2020). The reactions were either immediately terminated (time-point zero) or carried out for 15 min at 30 °C with agitation at 600 RPM in a PCMT Grant-bio Thermo-Shaker. Reaction termination was done by the addition of 5× SDS loading buffer (312.5 mM Tris-Cl pH 6.8, 10% SDS, 25% Glycerol, Bromophenol Blue) containing 20% β-mercaptoethanol. The samples were then boiled at 95 °C for 5 min, centrifuged at $1500 \times g$ for 1 min and analysed by SDS-PAGE and immunoblotting.

## *Salmonella* infections

*Salmonella enterica* serovar Typhimurium, strain 12023 (14028 s) and any isogenic mutants (Panagi et al, 2020) were used in this study. Strains carrying pWSK29 were grown with 50 µg/mL carbenicillin and 50 µg/mL kanamycin was included for bacterial mutant strains. 293ET cells were infected under SPI-1 inducing conditions. Bacteria were grown overnight in Luria broth (LB) and sub-cultured (1:33) in fresh LB for 3.5 h prior to infection of cells for 10 min at 37 °C with 10 µL of bacteria for cells seeded in 24-well plates. Following two PBS washes, the cells were incubated with 100 µg/mL gentamycin for 1–2 h at 37 °C and 20 µg/mL gentamycin for the remainder of the experiment.

## SDS-PAGE and immunoblotting

Whole-cell lysates, post-nuclear supernatant (PNS), pellet samples or in vitro kinase reaction samples were denatured by the addition of SDS-loading buffer and boiling at 95 °C for 5–7 min. Samples were then separated by SDS-PAGE using 10% or 14% polyacrylamide denaturing gels. Proteins were transferred to PVDF membrane (Millipore), blocked using 5% Bovine Serum Albumin (BSA) or 5% milk in TBS-T and incubated overnight with the appropriate primary antibody diluted in the same blocking solution. After three washes in Tris-buffered saline with 0.5% Tween-20 (TBS-T), membranes were incubated with the required HRP-conjugated secondary antibody for 1–2 h at RT, with rotation. Post-incubation, membranes were washed three times in TBS-T and proteins were visualised with ECL or ECL Plus detection reagents (Dako) on a Chemidoc™ Touch (Bio-Rad) or an iBright Imaging System Imaging System.

## Statistical analysis

Data were tested for statistical significance with GraphPad Prism software. The number of replicates for each experiment and the statistical test performed are indicated in the figure legend with P values are reported directly on the figure.

# Data availability

The mass spectrometry proteomics data have been deposited to the ProteomeXchange Consortium via the PRIDE partner repository with the dataset identifier PXD058368.

The source data of this paper are collected in the following database record: biostudies:S-SCDT-10_1038-S44319-025-00472-y.

# Peer review information

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

## Acknowledgements

We thank Prof David Holden and members of the Thurston laboratory for scientific discussion and reading of the manuscript and Laura Masino from the Structural Biology STP at the Francis Crick Institute for training and support on BLI experiments. Research reported in this publication was funded by a Biotechnology and Biological Sciences Research Council (BBSRC) David Phillips Fellowship (BB/R011834/1) awarded to TLMT, which also supported IP, an MRC DTP Studentship (MR/N014103/1) awarded to IP, a Medical Research Council (MRC) grant MR/V031058/1 awarded to TLMT and KR, which also funded JHM and IDDO, and an Engineering and Physical Sciences Research Council (EPSRC) grant EP/X02377X/1, underwriting European Research Council Starting Grant, Re-kin awarded to TLMT, which also supported IP. KR, TP and DE are supported by the Francis Crick Institute which receives its core funding from Cancer Research United Kingdom (CC 2075), the United Kingdom Medical Research Council (CC 2075), and the Wellcome Trust (CC 2075).

## Author contributions

**Ioanna Panagi**: Conceptualisation; Supervision; Investigation; Writing—original draft; Writing—review and editing. **Janina H Muench**: Investigation; Writing—review and editing. **Alexi Ronneau**: Investigation. **Ines Diaz-del-Olmo**: Investigation. **Agnel Aliyath**: Investigation. **Xiu-Jun Yu**: Investigation. **Hazel**

**Mak**: Investigation. **Enkai Jin**: Investigation. **Jingkun Zeng**: Investigation. **Diego Esposito**: Formal analysis; Investigation. **Elliott Jennings**: Investigation. **Timesh D Pillay**: Formal analysis; Investigation. **Regina A Günster**: Investigation. **Sarah L Maslen**: Methodology. **Katrin Rittinger**: Conceptualisation; Supervision; Funding acquisition; Writing—original draft; Writing—review and editing. **Teresa L M Thurston**: Conceptualisation; Supervision; Funding acquisition; Writing—original draft; Writing—review and editing.

Source data underlying figure panels in this paper may have individual authorship assigned. Where available, figure panel/source data authorship is listed in the following database record: biostudies:S-SCDT-10_1038-S44319-025-00472-y.

## Disclosure and competing interests statement

The authors declare no competing interests.

# Expanded View Figures

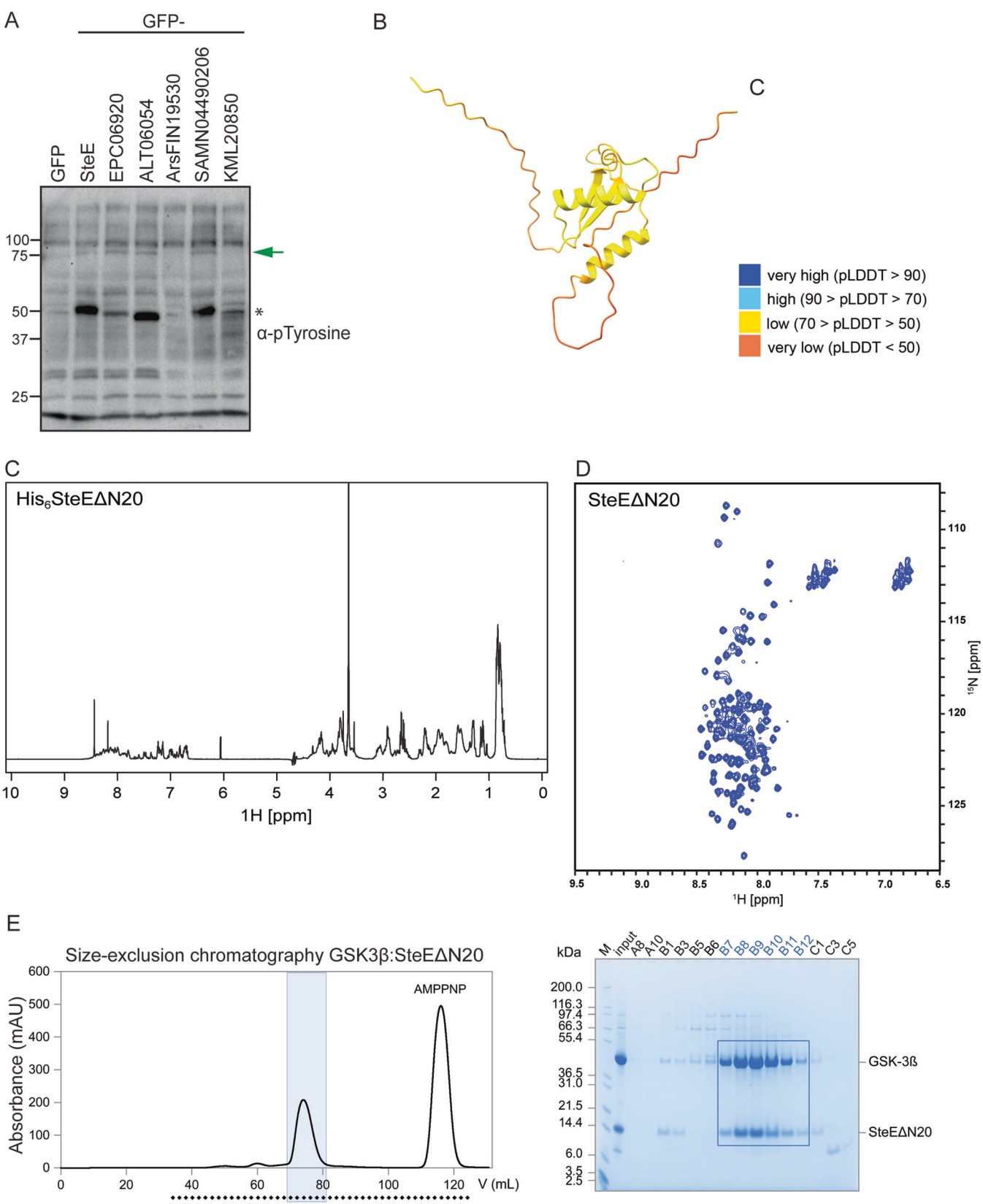

◀ **Figure EV1. SteE is a mainly disordered protein in solution.**

(A) 293ET cells expressing GFP or GFP-tagged SteE homologues were lysed and analysed by immunoblotting for tyrosine phosphorylation status. Data represent at least three independent biological repeats. * Indicates bands at the expected molecular weight of the GFP-tagged SteE homologues whilst the green arrow represents a band at the expected molecular weight of STAT3. (B) AlphaFold3 prediction of SteE (STM2585), coloured according to pLDDT values. (C) 1D $^1$H NMR spectrum of $^{15}$N-isotopically labelled His$_6$SteE $^{\Delta N20}$ at 266 μM in 18 mM Tris-HCl, 90 mM NaCl, 0.9 mM DTT, 0.9 mM MgCl$^{2+}$, 10% (v/v) D$_2$O, pH 7.0 was recorded at 700 MHz at 298 K. (D) 2D $^1$H,$^{15}$N-sofast HMQC spectrum of $^{15}$N-His$_6$SteE $^{\Delta N20}$ at 266 μM (same buffer as 1B), recorded at 700 MHz at 298 K. (E) Size-exclusion chromatography of SteE $^{\Delta N20}$ incubated with GSK3β. Data represent two independent biological repeats.

**SteE**

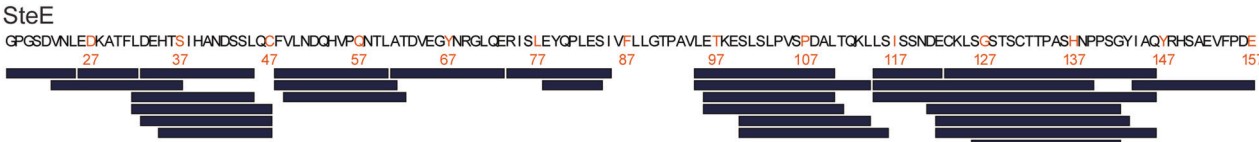

Total: 30 peptides, 93.6 % Coverage, 3.66 Redundancy

**GSK3β**

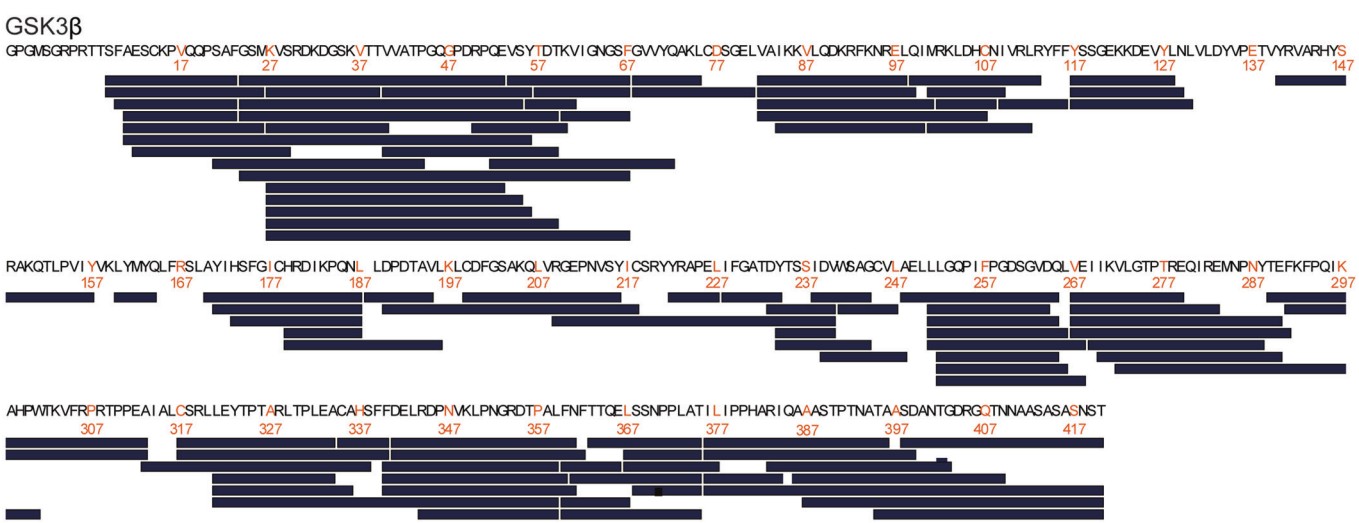

Total: 107 peptides, 93.6 % Coverage, 4.96 Redundancy

**Figure EV2. Peptide coverage for HDX-MS.**

Amino acid sequence and peptide coverage obtained for SteE and GSK3β. Related to Fig. 1E.

```
STM2585         --------------------------------MFTIN-STNRVAST-------IAP    16
KML20850        MVE----------FVYNELNTLRKGTALGSVWESNMFTIITNADRMA-A--------AAA    41
EPC06920        MLR---------SQRFL--SIPKLDYICDHVMEENMLAI-HNLNRI--E--------TAV    38
ALT06054        -----------------------------------------------------------     0
ArsFIN19530     -----------------------------------------------------------     0
SAMN04490206    MLSSRFDSVISSNQVLSEHTVEGAGSQRASVLSENTLSVQQTSRSMCKANDAGVDRQCIT    60

STM2585         YACVSDVNLEDKATFLDEHTSI-------HANDSSLQCFVLNDQHVPQNTLATDVEGYN-    68
KML20850        DTHLNHVNVEGKSGVVNLVVSVKNDPINMLRPDNFSQCVILNNLHVPQGALVTDIDNYN-   100
EPC06920        VKHVNCANVEGKPTFVASVSSVENVSTDMLKPKIFAQCVILNNLHVPQNIPDTDIEGYN-    97
ALT06054        ----------------------------MISNNMIKCNVFHDLHIPNDAAESDVETYK-    30
ArsFIN19530     --MSNKINITKDRPLIKNSQ-QQNQLKTNALISKIKNCFVLNNIHISTTIGSSKAKFYHQ    57
SAMN04490206    NACEVNAENNGVRVLMSALVSVGNVVSTGFVRQEINKVIVRDDIHVPQGSMPHDRKMCD-   119
                                            .   :  : .: *:        .   . .

STM2585         ------------------RGLQERISLEYQPLESIVFLLGTPAVLETKESLSLPVSPDAL   110
KML20850        ------------------KGLQLRINLEYNPKGSIVFLLGSPEALDANESLSLPIFSHVL   142
EPC06920        ------------------KGMQVRINQEYQPQGKTVFLLGSPEVLEPDESLSLPASPHIL   139
ALT06054        ------------------NDLSERISCEYHTDSNEVFILASPEELEDTESLSLAVSPKAL    72
ArsFIN19530     LMLRLNLDSSIQIKNYHHYHLTSRLNLEYNSTTENVALQGTPKNIHSAESLSLPVCPFLL   117
SAMN04490206    ------------------EGQRIRLNDEYHPLHHTVFLQGTPERLGIHHQLSLPVSPSML   161
                                  *:. **:       * : .:*  :    ..***       *

STM2585         TQKLLSISSNDECKLSGSTSCTTPASHNPPSGYIAQYRHSAEVFPDE-------------   157
KML20850        TQKLLNISNSKLCELSVKSNGYV------------------------------------   165
EPC06920        AQKLSSIANIKACAFSFESNGYVQRSE---DNFI--YRNGTS-LPL--------------   179
ALT06054        HQAISC------ELAKMTDHDLRERD-------MVETGKEIKPEEDVTKLHEYIIRAN   117
ArsFIN19530     SQKLIQVKNKFT----KKKKCKMKELD--------------------------------   140
SAMN04490206    TEKLIEVIREKNE-------KEQRASQAEKNGYVCQVGSVMTT----------------   197
                 : :

STM2585         ---    157
KML20850        ---    165
EPC06920        ---    179
ALT06054        GYV    120
ArsFIN19530     ---    140
SAMN04490206    ---    197
```

**Figure EV3.  Sequence alignment between SteE and its putative homologues.**

Amino acid sequence alignment between SteE (STM2585) and its putative homologues as generated by Clustal Omega. Conserved motifs within residues 95–116 of SteE are shown with shading. Known phosphosites in SteE are shown in bold. Asterisk (*) = fully conserved residue, Colon (:) = amino acid residues with similar properties, Period (.) = amino acid with weakly similar properties. Amino acid sequences are listed in Appendix Table S3.

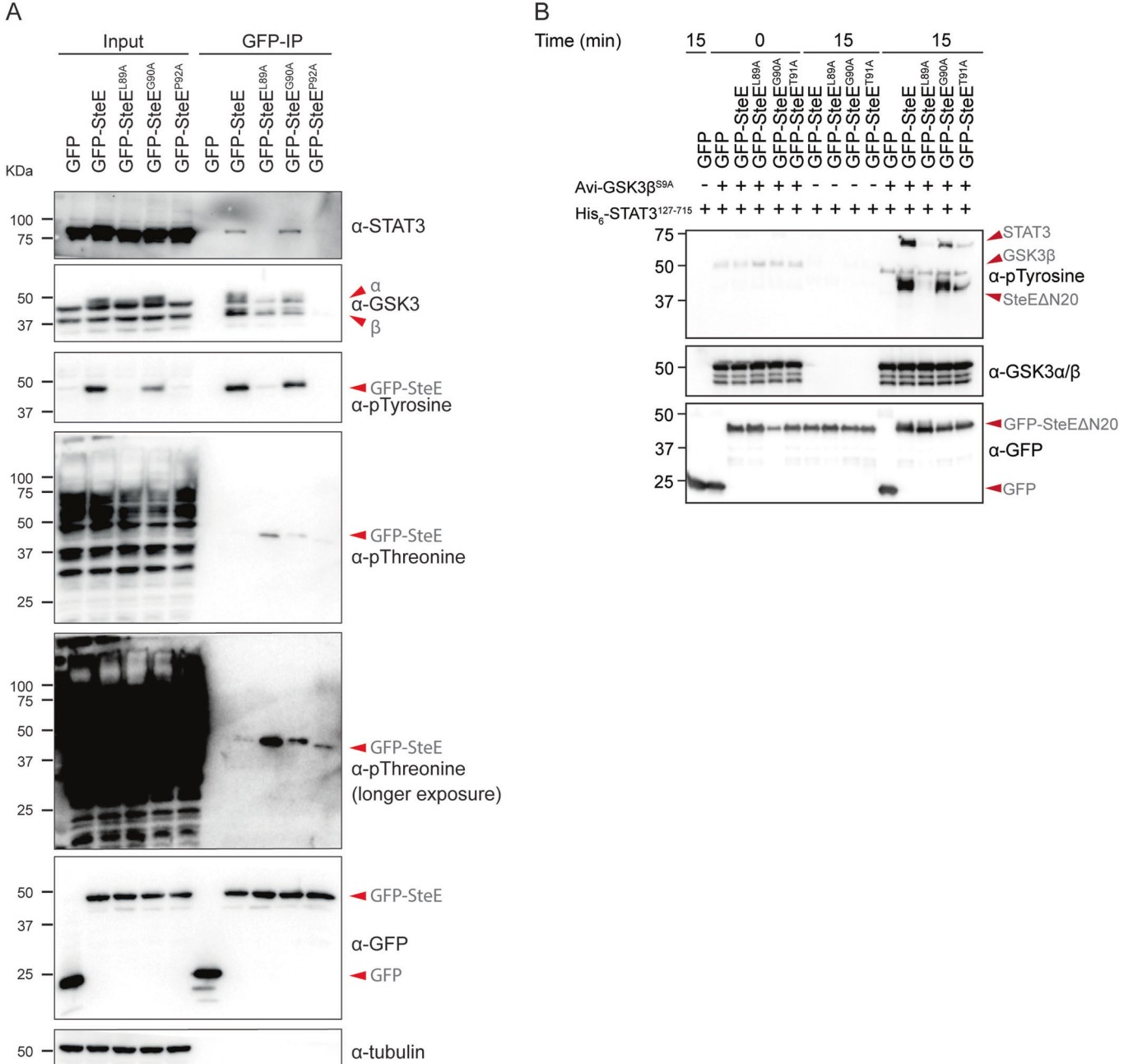

**Figure EV4.  Residue L89 of SteE is required for tyrosine phosphorylation mediated by GSK3.**

(A) 293ET cells expressing GFP or the indicated GFP-tagged SteE variant were lysed, and post-nuclear supernatants were subjected to GFP-TRAP immunoprecipitation (GFP:IP). Samples were analysed by immunoblotting. Data represent three independent biological repeats. (B) The indicated GFP-tagged proteins were immobilised on beads after expression from $GSK3\alpha/\beta^{-/-}$ 293ET cells. Tyrosine phosphorylation by GSK3β was analysed in an in vitro kinase reaction containing 1 mM ATP, 0.2 μM His$_6$-STAT3$^{127-715}$, with or without recombinant Avi-GSK3β$^{S9A}$ (0.2 μM). Data represent two biological repeats.

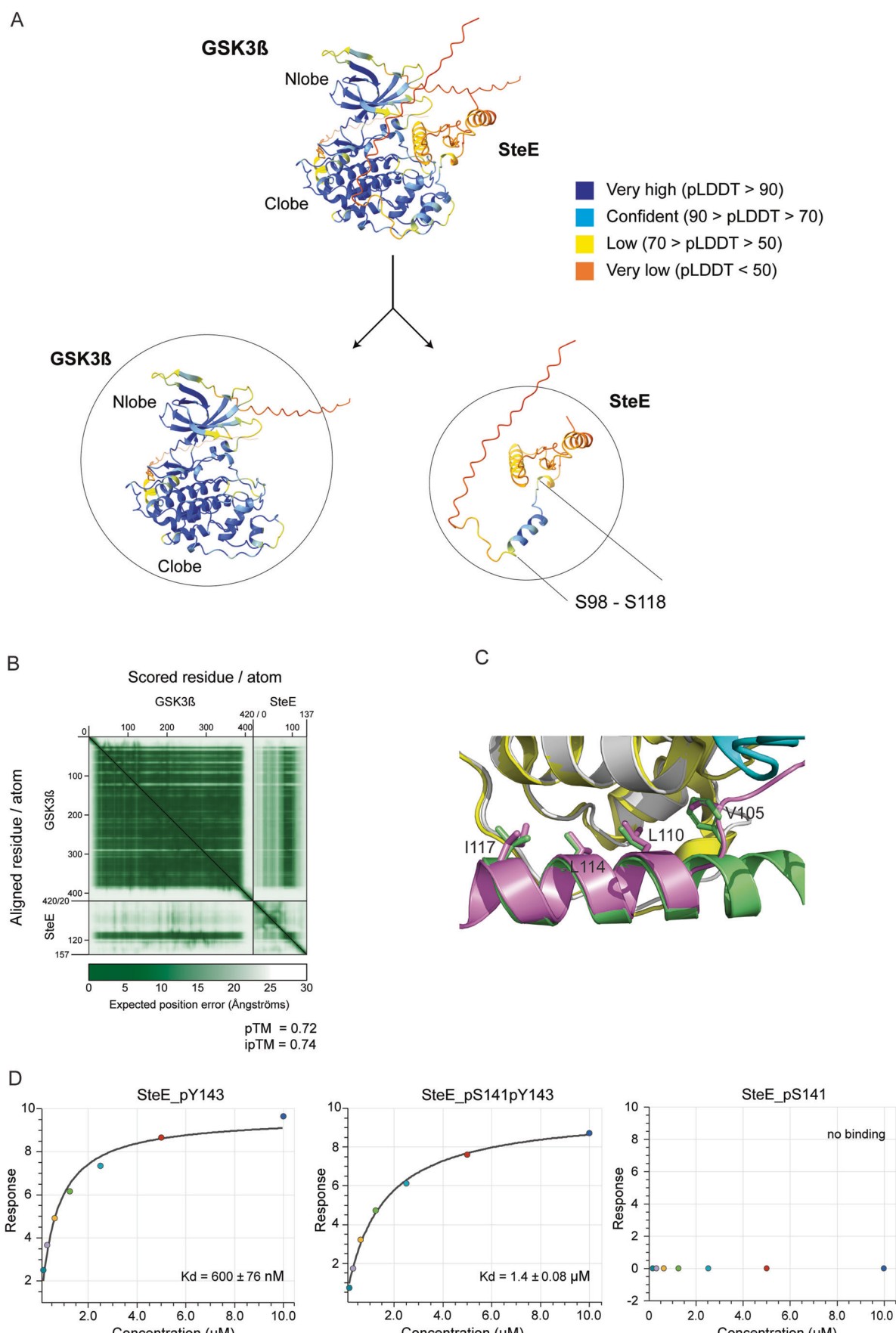

**Figure EV5.  Model of the GSK3β-SteE complex and role of SteE phosphorylation on Y143.**

(A) The polypeptide chains are coloured according to pLDDT values, shown in the same orientation as in Fig. 4B. For clarity, GSK3β and SteE are also shown separately to highlight regions on both proteins at various pLDDT confidence levels. (B) Expected position error matrix of the GSK3β-SteE complex AlphaFold3 prediction. Black lines indicate chain boundaries, with darker green regions showing higher confidence interactions. (C) Structure of GSK3β-axin peptide (yellow/green) superimposed with the predicted GSK3β-SteE interacting helix (magenta), with key hydrophobic interface residues of SteE highlighted. (D) Affinity of interaction between the indicated SteE[138-148] phosphorylated peptides and His$_6$STAT3[127-715] was determined by biolayer interferometry.

