## [Peer Review File · EMBO Reports]

Bacterial effectors mediate kinase reprogramming through mimicry of conserved eukaryotic motifs

Ioanna Panagi, Janina Muench, Alexi Ronneau, Ines Diaz-del-Olmo, Agnel Aliyath, Xiu-Jun Yu, Hazel Mak, Enkai Jin, Jingkun Zeng, Diego Esposito, Elliott Jennings, Timesh Pillay, Regina Günster, Sarah Maslen, Katrin Rittinger, and Teresa Thurston

Corresponding author(s): Teresa Thurston (Teresa.thurston@path.ox.ac.uk), Katrin Rittinger (katrin.rittinger@crick.ac.uk)

Review Timeline:

Transfer Date:	10th Oct 24
Editorial Decision:	10th Oct 24
Revision Received:	17th Mar 25
Editorial Decision:	8th Apr 25
Revision Received:	28th Apr 25
Accepted:	29th Apr 25

Editor: Achim Breiling

Transaction Report: This manuscript was transferred to EMBO reports following peer review at The EMBO Journal.

Dear Dr. Thurston,

Thank you for transferring your manuscript to EMBO reports. I now went through the manuscript, the referee reports from The EMBO Journal (attached again below) and your point-by-point-response (revision plan). The referees have several concerns and suggestions to improve the manuscript, or to strengthen the data and the conclusions drawn. Going through your revision plan, it seems that the referee concerns will be adequately addressed.

I would like to invite you to revise your manuscript with the understanding that all concerns of the referees must be addressed in the revised manuscript as indicated in your revision plan. Acceptance of your manuscript will depend on a positive outcome of another round of review at EMBO reports, using the same referees.

EMBO Reports emphasises novel functional insight with clear in vivo relevance over detailed mechanistic insight. Thus, EMBO Reports will not require addressing points regarding more mechanism experimentally. However, it will be necessary that during the revision you address all points questioning the main conclusions of the study, and all technical concerns, or points regarding the experimental designs, model systems used, or data presentation.

1) a .docx formatted version of the final manuscript text (including legends for main figures, EV figures and tables), but without the figures included. Please make sure that changes are highlighted to be clearly visible. Figure legends should be compiled at the end of the manuscript text.

2) individual production quality figure files as .eps, .tif, .jpg (one file per figure), of main figures and EV figures. Please upload these as separate, individual files upon re-submission. Please make sure that all figure panels are called out separately and sequentially in the manuscript text

For more details please refer to our guide to authors:

See also our guide for figure preparation:

Moreover, please consult our guidelines for figure legend preparation:

4) a complete author checklist, which you can download from our author guidelines

(<https://www.embopress.org/page/journal/14693178/authorguide>). Please insert page numbers in the checklist to indicate where the requested information can be found in the manuscript. The completed author checklist will also be part of the RPF.

5) that primary datasets produced in this study (e.g. RNA-seq, ChIP-seq and array data) are deposited in an appropriate public database. This is now mandatory (like the COI statement). If no primary datasets have been deposited in any database, please state this in this section (e.g. 'No primary datasets have been generated and deposited').

The accession numbers and database should be listed in a formal "Data Availability " section (placed after Materials & Methods) that follows the model below. Please note that the Data Availability Section is restricted to new primary data that are part of this study.

Data availability

8) Regarding data quantification and statistics, please make sure that the number "n" for how many independent experiments were performed, their nature (biological versus technical replicates), the bars and error bars (e.g. SEM, SD) and the test used to calculate p-values is indicated in the respective figure legends (also for potential EV figures and all those in the final Appendix). Please also check that all the p-values are explained in the legend, and that these fit to those shown in the figure. Please provide statistical testing where applicable. Please avoid the phrase 'independent experiment', but clearly state if these were biological or technical replicates. Please also indicate (e.g. with n.s.) if testing was performed, but the differences are not significant. In case n=2, please show the data as separate datapoints without error bars and statistics.

See also:

<http://www.embopress.org/page/journal/14693178/authorguide#statisticalanalysis>

9) Please add scale bars of similar style and thickness to microscopic images, using clearly visible black or white bars (depending on the background). Please place these in the lower right corner of the images themselves. Please do not write on or near the bars in the image but define the size in the respective figure legend.

10) Please note our reference format:

11) We updated our journal's competing interests policy in January 2022 and request authors to consider both actual and perceived competing interests. Please review the policy <https://www.embopress.org/competing-interests> and add a statement declaring your competing interests. Please name that section 'Disclosure and Competing Interests Statement' and add it after the author contributions section.

12) Please order the sections like this using these names:

Title page - Abstract - Keywords - Introduction - Results - Discussion - Methods - Data availability section (DAS) -

Acknowledgements (including funding information) - Disclosure and Competing Interests Statement - References - Figure legends - Expanded View Figure legends

13) Please make sure that all the funding information is also entered into the online submission system and is complete and similar to the one in the manuscript text file (in the Acknowledgements).

14) We now use CRediT to specify the contributions of each author in the journal submission system. CRediT replaces the author contribution section. Please use the free text box to provide more detailed descriptions. Thus, please do not provide your final manuscript text file with an author contributions section. See also guide to authors:
<https://www.embopress.org/page/journal/14693178/authorguide#authorshipguidelines>

15) All Materials and Methods need to be described in the main text using our 'Structured Methods' format, which is required for all research articles. According to this format, the Methods section should include a Reagents and Tools Table (listing key reagents, experimental models, software, and relevant equipment and including their sources and relevant identifiers), uploaded as separate file, followed by a Methods section in which we encourage the authors to describe their methods using a step-by-step protocol format with bullet points, to facilitate the adoption of the methodologies across labs. More information on how to adhere to this format as well as downloadable templates (.doc or .xls) for the Reagents and Tools Table can be found in our author guidelines (section 'Structured Methods'):

I look forward to seeing a revised form of your manuscript when it is ready.

Best,

Referee #1:

This manuscript focuses on the *Salmonella enterica* type III effector, SteE (also known as SarA and GogC). Work primarily from two labs, Teresa Thurston and Dennis Ko, has shown that SteE does not have enzymatic activity. Rather, SteE binds to glycogen synthase kinase (GSK), leading to its phosphorylation at Thr91, Ser141 and Tyr143 residues. The latter modification directs SteE binding of STAT3, recruiting it to the SteE-GSK complex, where it is then phosphorylated by GSK, directing it to the nucleus and thereby initiating a STAT3-dependent anti-inflammatory transcriptional program. This manuscript by Panagi and others seeks to identify the molecular interactions between SteE and GSK, and whether SteE homologs have similar effects on GSK i.e. converting GSK from a serine/threonine kinase to a serine/threonine/tyrosine kinase. The biochemical and structure-function studies are rigorous and the conclusions drawn from the results are justified. The figures are well-presented. The weak link is the conservation studies; there is no evidence that these SteE orthologs are secreted by these bacteria (most of which I have never heard of before). Without delivery into a host cell, these SteE orthologs would not be able to interact with GSK, providing a very poor rationale for studying their ability to modulate GSK phosphorylation activity. Additionally, another group has already published the requirement for host phosphorylation of SteE/SarA and virulence, and STAT3-YXXQ-containing peptide binding data (Gibbs et al., 2020), which negatively affects the impact of the results presented here.

(1) SteE homolog studies. The five Gram-negative bacteria that these homologs are found in - *Erwinia pyrifoliae*, *Edwardsiella ictaluri*, *Arsenophonus nasoniae*, *Pseudomonas umsongensis* and *Leclercia adecarboxylata* - are vastly understudied and, I

would argue, little to no evidence exists that these bacteria can inject/translocate/deliver their own proteins into a host cell. As stated above, this is essential for these "homologs" to function like SteE. Maybe the authors are aware of studies they can cite which show their ability to deliver proteins? Do they encode for secretion systems? Type III? Type IV? The authors should really solidify their argument for these studies.

(2) In relation to the above point, remove statements such as "having established that reprogramming of GSK by bacterial effectors is evolutionarily conserved..."

(3) Figure 4A. The amino acid numbering is inconsistent between the Thurston group and Ko group for the tyrosine phosphorylation site (YXXQ motif) in SteE/SarA. I am not sure why. Ko has it as Tyr167 (Gibbs et al., 2020) and Thurston has it Tyr143. Regardless, Ko's group has already shown that phosphorylation at this motif is required for the virulence phenotype of SteE in mice (see Figure 2 in Gibbs et al., 2020). Please cite.

(4) Figure S5. Similarly, the Ko group has already reported the binding affinities of STAT3 to YXXQ-containing peptides from SarA/SteA (see Figure 3 in Gibbs et al., 2020). Please cite.

(5) I would recommend the "molecular mimicry" angle be toned down. Crystallography studies from many years ago revealed that bacterial effectors adopt similar structures to host cell proteins to intercept host cell functions i.e. this is not a new concept in bacteria-host cell interactions.

Referee #2:

The manuscript by Panagi and colleagues is well written and clear throughout, with properly controlled experiments. It aims to investigate a sequence-related series of bacterial effector proteins related to Salmonella SteE. A fair bit is already known about the broad effects of SteE on GSK3, including the finding that SteE and the GSK3 transcriptional substrate STAT3 are both tyrosine phosphorylated in its presence, broadening the original model that GSK3 Tyr phosphorylation was 'simply' co-translational/maturational. The work reported here reveals convincing molecular interactions and phosphorylation motifs between SteE and GSK3 (and a group of other SteE-like proteins from microbes, but not including the quite close homologue KML20850). A near-comprehensive set of mutational swaps leads the authors to conclude that an SLPLVSP motif is a SLiM that targets these proteins to GSK3 (and this precise sequence can also be transplanted onto KML20850 to make it do so, which is a nice experiment). Further analysis demonstrates S/T/Y sites on SteE (it is both a binding partner and substrate) and show that T91 in SteE dictates complex formation and GSK3 Tyr phosphorylation of SteE and STAT3. The LGTP motif (notably Leu and Pro) of SteE (shared with GSK3) is also relevant for driving STAT3 Tyr phosphorylation. Finally, there is clear evidence presented through changing SteE L89 (compared as L89A vs. a control Y143F) that SteE is important for virulence in vivo (mouse model).

The paper is exciting overall, and there is a nice selection of biophysical/prediction, cellular and in vivo data. I have only one major comment:

1) Leading from the discussion, if reprogramming of GSK3 into a Tyr-directed kinase is to be a key outcome of co-expression/interaction with GSK3 (as shown in the paper through pGSK3 and pSTAT3) by a broad variety of SteE proteins, then what does a total pTyr blot show for such co-expression cell lysates? Is there a ladder of proteins? Even better (and can be done at Imperial/Crick) what does phosphoproteomics through Mass Spectrometry indicate are the sites of Tyr that are induced by, say, SteE, and are they different from EPC06920, ALT0606054, but not KML20850?

Minor point:

1) Please number pages and lines. It is stressful to review when this hasn't been done.

2) Supplementary Figure 4 should be in Figure 4

3) What does 'recipe' mean Under SDS-PAGE and Immunoblotting. 'After three washes in TBS-T (recipe), membranes...'

Referee #3:

This manuscript by Panagi and co-workers investigates how the Salmonella virulence factor SteE and a set of its homolog from other bacterial species act to "reprogram" the kinase activity of GSK3beta, apparently promoting its activity as a Tyr kinase and scaffolding GSK3beta to Tyr phosphorylate host STAT3. They argue that SteE accomplishes this through several short linear sequence motifs that mimic those of eukaryotic proteins. Using HDX-MS to identify putative interaction surfaces between SteE and GSK3b, they home in on two regions of SteE harboring linear motifs, examining a series of point mutants for their ability to accumulate in host cells, bind GSK3b and STAT3, and induce STAT3 Tyr phosphorylation. One sequence mimics the canonical GSK3b recognition motif (S-x-x-x-pS) in its central region, though important features of this sequence do not correspond to eukaryotic motifs. The authors do show strikingly that a SteE homolog that does not bind GSK3b can be made to interact by installing a key portion of this motif. The second region is an LGTP sequence in SteE that is suggested to mimic the L/x-G-x-P motif in GSK3b itself known to be important for GSK3b autophosphorylation on a Tyr residue essential for its activity. Using AlphaFold multimer, they build a model of the GSK3b-SteE complex that is consistent with their HDX-MS and mutagenesis experiments. Overall this study provides valuable insight into how a bacterial virulence factor co-opts host signaling. However, I'm not entirely sold on aspects of the model, and I feel that the manuscript should be rewritten (and the title changed) so as to back off on the strong claims of motif mimicry.

Main comments:

1. Prior research from another group had identified an L/xGxP motif in the catalytic domain of GSK3beta and other CMGC group kinases as essential for Tyr autophosphorylation, which is thought to occur in cis on a folding intermediate. Here, the authors suggest that the same motif present in SteE (LGTP) mimics this sequence and reprograms GSK3beta into a Tyr kinase. While this region of SteE does appear important for interacting with GSK3beta and possibly inducing its Tyr kinase activity, its putative binding site in the authors' AlphaFold model appears quite distant from the native L/xGxP sequence in GSK3beta. From this standpoint it is hard to understand how the SteE LGTP sequence could be mimicking the endogenous sequence, and in all likelihood the sequence similarity is coincidental. While I do think the present study has value in proposing a structural model for how SteE acts as a scaffold, I don't think the argument that this reflects host protein mimicry is convincing.
2. Whether the LGTP motif of SteE indeed induces GSK3 Tyr kinase activity, as opposed to being an essential part of the GSK3b binding interface, is based on the experiments shown in Figs 3F and 3G. Collectively these experiments show that SteE-L89A can still bind GSK3b in a pulldown experiment, yet fails to efficiently induce Tyr phosphorylation of itself (or STAT3, but this is presumed to require SteE Tyr phosphorylation for recruitment). However, the L89A mutation appears to weaken the interaction with GSK3b. Conducting the experiment in Fig 3G in a more quantitative manner - for example examining the potency of SteE (EC50 and Emax) and various mutants in activating GSK3b Tyr kinase activity could shed some light on this, as the prediction would be that increasing concentrations of SteE-L89A would not be able to overcome the defect in inducing Tyr kinase activity. It would also be important to examine whether this motif is important in inducing the canonical Ser/Thr kinase activity of GSK3b - perhaps the LGTP sequence simply acts to recruit the kinase to SteE or acts as a GSK3b activator rather than truly "reprogramming" the kinase.
3. The experiment shown in Fig 3G is key as it demonstrates that Leu89 and Pro91 of SteE are essential for inducing GSK3b Tyr kinase activity. However the experiment is missing controls in which GSK3b was left out of the reaction. This is important to ensure the observed kinase activity was not due to contamination from SteE preparations isolated from 293 cells (I realize these were GSK3a/b knockout cells, but it doesn't rule out other Tyr kinases co-purifying with SteE).
4. Fig 4B showing the AlphaFold multimer model would benefit from zoomed in views illustrating the presumed interaction interface, which is not well-represented in the current version of this panel. The main text needs to elaborate on the model. There is a generic statement that it is consistent with the HDX-MS data, but no details are given, and aside from a few highlighted residues, it's not clear what residues form the interface (like the helix whose binding site spans the GSK3b N- and C-lobes). The ipTM score plots for the AlphaFold multimer model in Fig 4B should also be shown to allow others to assess the confidence in the model, and it would be useful to include a PDB file. In the model, GSK3b would not be Tyr phosphorylated - does it capture the catalytic site conformation in reported crystal structures?
5. The AlphaFold model predicts multiple points of interaction beyond the short motifs identified by sequence alignment, but the validity of this model was not tested. For example, the similarity of the binding mode of a predicted helical region of SteE in the model to that of the GSK3b scaffold Axin is interesting. In Axin this is an amphipathic helix that makes extensive hydrophobic contacts with the kinase. Are the hydrophobic residues conserved to SteE? Does mutation of these residues disrupt the interaction with and Tyr phosphorylation of GSK3b?

Minor comments:

1. Can the authors speculate as to why Thr91 is required for SteE to interact with GSK3b? The authors suggest that it must be phosphorylated, but is it possible that non-phosphorylated Thr91 will also interact? A T91V substitution or dephosphorylating SteE with phosphatase treatment could address this question.
2. HDX-MS panels in Fig 1 don't have X-axis labels. I gather each datapoint is a distinct peptide species shown in Fig S2.

Panagi et al., pbp response

Referee #1:

This manuscript focuses on the *Salmonella enterica* type III effector, SteE (also known as SarA and GogC). Work primarily from two labs, Teresa Thurston and Dennis Ko, has shown that SteE does not have enzymatic activity. Rather, SteE binds to glycogen synthase kinase (GSK), leading to its phosphorylation at Thr91, Ser141 and Tyr143 residues. The latter modification directs SteE binding of STAT3, recruiting it to the SteE-GSK complex, where it is then phosphorylated by GSK, directing it to the nucleus and thereby initiating a STAT3-dependent anti-inflammatory transcriptional program. This manuscript by Panagi and others seeks to identify the molecular interactions between SteE and GSK, and whether SteE homologs have similar effects on GSK i.e. converting GSK from a serine/threonine kinase to a serine/threonine/tyrosine kinase.

The biochemical and structure-function studies are rigorous and the conclusions drawn from the results are justified. The figures are well-presented.

We thank this reviewer for their positive assessment of the data presented in our manuscript.

The weak link is the conservation studies; there is no evidence that these SteE orthologs are secreted by these bacteria (most of which I have never heard of before). Without delivery into a host cell, these SteE orthologs would not be able to interact with GSK, providing a very poor rationale for studying their ability to modulate GSK phosphorylation activity.

Please find our detailed response below.

Additionally, another group has already published the requirement for host phosphorylation of SteE/SarA and virulence, and STAT3-YXXQ-containing peptide binding data (Gibbs et al., 2020), which negatively affects the impact of the results presented here.

As noted by the reviewer, phosphorylation of Y143 representing a YXXQ has been published, and we reference this study several times throughout the manuscript.

Importantly, however, previous studies did not investigate the specific function of T91, did not test the importance of S141 phosphorylation and did not test whether Y phosphorylation (within the YXXQ motif) might depend on preceding S/T phosphorylation. Therefore, when considering the mechanism of GSK3 reprogramming by SteE in this manuscript, we find it important that we independently analysed the importance of Y143 phosphorylation, including in relation to T91 and S141 phosphorylation. Instead of negatively impacting our work, we believe this places prior findings into the context so that we can propose a model for the requirements of kinase reprogramming.

(1) SteE homolog studies. The five Gram-negative bacteria that these homologs are found in - *Erwinia pyrifoliae*, *Edwardsiella ictaluri*, *Arsenophonus nasoniae*, *Pseudomonas umsongensis* and *Leclercia adecarboxylata* - are vastly understudied and, I would argue, little to no evidence exists that these bacteria can inject/translocate/deliver their own proteins into a host cell. As stated above, this is essential for these "homologs" to function like SteE. Maybe the authors are aware of studies they can cite which show their ability to deliver proteins? Do they encode for secretion systems? Type III? Type IV? The authors should really solidify their argument for these studies.

We apologise that we did not include sufficient references to unequivocally demonstrate the presence of T3SS apparatuses, as well as the presence of other

conserved effectors with importance to pathogen virulence. We have revised the text [starting on line 110] to include the following:

Interestingly, even though little is known about *Pseudomonas umsongensis*, *Erwinia pyrifoliae*, *Edwardsiella ictaluri* and *Arsenophonus nasoniae* each encode at least one T3SS showing homology to the injectisomes present in *Salmonella*. The genome of the plant associated pathogen *Erwinia pyrifoliae*, contains two distinct T3SSs: a conserved *hrp1* gene cluster with associated effectors and enzymes involved in systemic virulence and a second T3SS that is related to the SPI-1 encoded *inv/spa* cluster of *Salmonella* Typhimurium (Oh *et al*, 2005; Smits *et al*, 2010). *Edwardsiella ictaluri* represents a human and fish pathogen of concern in the food chain. The genome encodes for a Ssa-Esc like T3SS that resembles the SPI-2 encoded *Salmonella* T3SS and this injectisome is essential for intracellular replication and virulence of the pathogen (Thune *et al*, 2007; Dubytska *et al*, 2016). *Arsenophonus nasoniae* is a heritable microbe associated with male-killing in parasitic wasps across Europe (Nadal-Jimenez *et al*, 2023). These bacteria encode two complete T3SSs, with the second resembling the SPI-1 encoded *inv/spa*-like apparatus of *Salmonella*. Furthermore, open reading frames, resembling at least 10 effectors from *Salmonella* or other Gram-negative bacteria have been identified in the *A. nasoniae* genome (Wilkes *et al*, 2010; Siozios *et al*, 2023). Effector homologues have also been identified in other pathogens including a homologue of SopA in *Erwinia* (Kube *et al*, 2010) and a homologue of SpvC in *E. ictaluri*, which is required for virulence (Dubytska *et al*, 2022). The presence of both a T3SS injectisome and a putative SteE homologue in these pathogens raises the intriguing possibility that kinase reprogramming represents a more general virulence mechanism.

(2) In relation to the above point, remove statements such as "having established that reprogramming of GSK by bacterial effectors is evolutionarily conserved...".

Given the strong evidence that these bacteria do indeed have the required T3SS injectisomes to translocate effectors, we believe we are justified in the above statement. However, we have made the following adjustment to tone down the statement so that it reads "Given that reprogramming of human GSK3 by diverse bacterial proteins was observed..." [line 145].

(3) Figure 4A. The amino acid numbering is inconsistent between the Thurston group and Ko group for the tyrosine phosphorylation site (YXXQ motif) in SteE/SarA. I am not sure why. Ko has it as Tyr167 (Gibbs *et al.*, 2020) and Thurston has it Tyr143. Regardless, Ko's group has already shown that phosphorylation at this motif is required for the virulence phenotype of SteE in mice (see Figure 2 in Gibbs *et al.*, 2020). Please cite.

We have previously shown in Panagi *et al.*, 2020, that SteE/SarA was incorrectly annotated in the databases and demonstrated that the 157 amino acid sequence we use is a) sufficient for the phenotypes observed and b) produced by a consensus Shine Delgarno sequence that corresponds with transcriptomics data. As we appreciate this is confusing to readers unfamiliar with our prior study, we have added this sentence "Of note, Y143 of SteE is equivalent to residue Y167 analysed in Gibbs *et al.*,⁵ with the difference in amino acid numbering arising from a miss-annotation of the start methionine of STM2585 (SteE/SarA)⁴." [line 72 of the revised manuscript].

We now include specific mention of the analysis of the Y143 mutant on line 72 and 300 with citation to Gibbs *et al.*, and have modified our text to reflect why we have carried out our *in vivo* experiment.

(4) Figure S5. Similarly, the Ko group has already reported the binding affinities of STAT3 to YXXQ-containing peptides from SarA/SteA (see Figure 3 in Gibbs et al., 2020). Please cite.

We apologise for this omission and have added the Gibbs et al reference to line 308 of the manuscript prior to referencing our new work in Figure S5 that compares the requirement of S141 and Y143 phosphorylation.

(5) I would recommend the "molecular mimicry" angle be toned down. Crystallography studies from many years ago revealed that bacterial effectors adopt similar structures to host cell proteins to intercept host cell functions i.e. this is not a new concept in bacteria-host cell interactions.

We are sorry if we have not expressed this clearly: we are not claiming to have invented molecular mimicry as a new concept. Instead, we wish to highlight that this largely intrinsically disordered protein consists of several short linear motifs that appear to mimic eukaryotic motifs. We have made several changes that tone down this angle [lines 37-41, 88, 178, 213, 293, 333], whilst still proposing that SteE represents a new example that exploits the molecular mimicry of Short linear motifs.

Referee #2:

The manuscript by Panagi and colleagues is well written and clear throughout, with properly controlled experiments. It aims to investigate a sequence-related series of bacterial effector proteins related to Salmonella SteE. A fair bit is already known about the broad effects of SteE on GSK3, including the finding that SteE and the GSK3 transcriptional substrate STAT3 are both tyrosine phosphorylated in its presence, broadening the original model that GSK3 Tyr phosphorylation was 'simply' co-translational/maturational. The work reported here reveals convincing molecular interactions and phosphorylation motifs between SteE and GSK3 (and a group of other SteE-like proteins from microbes, but not including the quite close homologue KML20850). A near-comprehensive set of mutational swaps leads the authors to conclude that an SLSPVSP motif is a SLiM that targets these proteins to GSK3 (and this precise sequence can also be transplanted onto KML20850 to make it do so, which is a nice experiment). Further analysis demonstrates S/T/Y sites on SteE (it is both a binding partner and substrate) and show that T91 in SteE dictates complex formation and GSK3 Tyr phosphorylation of SteE and STAT3. The LGTP motif (notably Leu and Pro) of SteE (shared with GSK3) is also relevant for driving STAT3 Tyr phosphorylation. Finally, there is clear evidence presented through changing SteE L89 (compared as L89A vs. a control Y143F) that steE is important for virulence in vivo (mouse model).

We thank this reviewer for their positive and supportive assessment of our manuscript.

The paper is exciting overall, and there is a nice selection of biophysical/prediction, cellular and in vivo data. I have only one major comment:

1) Leading from the discussion, if reprogramming of GSK3 into a Tyr-directed kinase is be a key outcome of co-expression/interaction with GSK3 (as shown in the paper through pGSK3 and pSTAT3) by a broad variety of SteE proteins, then what does a total pTyr blot show for such co-expression cell lysates? Is there a ladder of proteins? Even better (and can be done at Imperial/Crick) what does phosphoproteomics through Mass Spectrometry indicate are

the sites of Tyr that are induced by, say, SteE, and are they different from EPC06920, ALT0606054, but not KML20850?

Carrying out new phosphoproteomic experiments on SteE from multiple species would go beyond the scope of this study, however, we have included a total pTyr blot on cell lysates as requested by the reviewer. This data, included as **new Figure EV1A** and reproduced below for ease, shows bands corresponding to the molecular weight of STAT3 at approximately 75 kDa (green arrow), as well as the GFP-tagged SteE putative homologues at approximately 50 kDa (*).

Minor point:

1) Please number pages and lines. It is stressful to review when this hasn't been done.

This has been done.

2) Supplementary Figure 4 should be in Figure 4

We agree with the reviewer and have moved the data so that it is now Figure 4C.

3) What does 'recipe' mean Under SDS-PAGE and Immunoblotting. 'After three washes in TBS-T (recipe), membranes...'

This was an error, and we have corrected this mistake [line 750] to include the recipe "Tris-buffered saline with 0.5% Tween-20".

Referee #3:

This manuscript by Panagi and co-workers investigates how the Salmonella virulence factor SteE and a set of its homolog from other bacterial species act to "reprogram" the kinase activity of GSK3beta, apparently promoting its activity as a Tyr kinase and scaffolding GSK3beta to Tyr phosphorylate host STAT3. They argue that SteE accomplishes this through several short linear sequence motifs that mimic those of eukaryotic proteins. Using HDX-MS to identify putative interaction surfaces between SteE and GSK3b, they home in on two regions of SteE harboring linear motifs, examining a series of point mutants for their

ability to accumulate in host cells, bind GSK3b and STAT3, and induce STAT3 Tyr phosphorylation. One sequence mimics the canonical GSK3b recognition motif (S-x-x-x-pS) in its central region, though important features of this sequence do not correspond to eukaryotic motifs. The authors do show strikingly that a SteE homolog that does not bind GSK3b can be made to interact by installing a key portion of this motif. The second region is an LGTP sequence in SteE that is suggested to mimic the L/x-G-x-P motif in GSK3b itself known to be important for GSK3b autophosphorylation on a Tyr residue essential for its activity. Using AlphaFold multimer, they build a model of the GSK3b-SteE complex that is consistent with their HDX-MS and mutagenesis experiments. Overall this study provides valuable insight into how a bacterial virulence factor co-opts host signaling. However, I'm not entirely sold on aspects of the model, and I feel that the manuscript should be rewritten (and the title changed) so as to back off on the strong claims of motif mimicry.

We thank this reviewer for highlighting the strengths and insights of our study.

Main comments:

1. Prior research from another group had identified an L/xGxP motif in the catalytic domain of GSK3beta and other CMGC group kinases as essential for Tyr autophosphorylation, which is thought to occur in cis on a folding intermediate. Here, the authors suggest that the same motif present in SteE (LGTP) mimics this sequence and reprograms GSK3beta into a Tyr kinase. While this region of SteE does appear important for interacting with GSK3beta and possibly inducing its Tyr kinase activity, its putative binding site in the authors' AlphaFold model appears quite distant from the native L/xGxP sequence in GSK3beta. From this standpoint it is hard to understand how the SteE LGTP sequence could be mimicking the endogenous sequence, and in all likelihood the sequence similarity is coincidental. While I do think the present study has value in proposing a structural model for how SteE acts as a scaffold, I don't think the argument that this reflects host protein mimicry is convincing.

As noted by the reviewer, auto-tyrosine phosphorylation by GSK3 is thought to occur on a folding intermediate, therefore, the location of the LGTP motif in the fully folded conformation may not be representative of the prone-to-autophosphorylate intermediate. We state in the discussion [line 387], "Whether the LGTP motif of SteE might enable GSK3 to re-adopt the "prone-to-autophosphorylate" through protein conformational changes requires further investigation.", acknowledging that further experimentation is required. Then, as the AlphaFold model shown in Figure 4B, is of the initial unphosphorylated complex, we can only speculate whether phosphorylation of SteE causes GSK3 to adopt a different conformation. To capture this clearly in our discussion we have added that further investigation "should include an analysis of phosphorylated SteE intermediates."

2. Whether the LGTP motif of SteE indeed induces GSK3 Tyr kinase activity, as opposed to being an essential part of the GSK3b binding interface, is based on the experiments shown in Figs 3F and 3G. Collectively these experiments show that SteE-L89A can still bind GSK3b in a pulldown experiment, yet fails to efficiently induce Tyr phosphorylation of itself (or STAT3, but this is presumed to require SteE Tyr phosphorylation for recruitment). However, the L89A mutation appears to weaken the interaction with GSK3b.

Conducting the experiment in Fig 3G in a more quantitative manner - for example examining the potency of SteE (EC50 and Emax) and various mutants in activating GSK3b Tyr kinase activity could shed some light on this, as the prediction would be that increasing concentrations of SteE-L89A would not be able to overcome the defect in inducing Tyr kinase activity. It would also be important to examine whether this motif is important in inducing the canonical Ser/Thr kinase activity of GSK3b - perhaps the LGTP sequence

simply acts to recruit the kinase to SteE or acts as a GSK3b activator rather than truly "reprogramming" the kinase.

The reviewer raises an important consideration regarding interaction of L89A with GSK3b. To test this further, we analysed the interaction between SteE^{L89A} and GSK3 during the physiological conditions of infection. Unlike, SteE^{T91A} and SteE^{P92A}, which do not stably interact with GSK3 and which are unstable following their translocation from bacteria (see Figures 3A,B and 3E,F), SteE^{L89A} was stable and interacted with GSK3 β (**new Figure 3H**). As seen in transfection, a reduction in interaction with GSK3 α was apparent (upper band in GSK3 blot). We included these findings in the revised manuscript [lines286-293] and summarise that whereas mutation of L89 results in a decreased stable interaction with GSK3 α , the more profound phenotype is the complete loss of Y phosphorylation observed during infection. Furthermore, as SteE functions via either GSK3 α or GSK3 β (Panagi *et al*, 2020) this reduction in interaction for GSK3 α is unlikely to explain the complete loss of tyrosine phosphorylation observed for SteE^{L89A}.

Figure 3H

Cells infected with the indicated *Salmonella* strains for 17 h were lysed, and post-nuclear supernatants were subjected to HA-TRAP immunoprecipitation (HA:IP). Input and IP samples were analysed by immunoblotting. NI – non-infected. Data represent three independent biological repeats.

Then, as suggested by the reviewer, we have analysed threonine phosphorylation upon transfection of SteE^{L89A}. This new data included as **new Figure EV4A** revealed that mutation of L89 does not impact the ability of GSK3 to mediate threonine phosphorylation of SteE. This supports the hypothesis that L89 represents an important residue within the LGTP motif that is required for inducing the tyrosyl-directed phosphorylation activity of GSK3. That this motif resembles a motif in GSK3, required for tyrosine phosphorylation, is striking and worthy of discussion.

Figure EV4A

293ET cells expressing GFP or the indicated GFP-tagged SteE variant were lysed, and post-nuclear supernatants were subjected to GFP-TRAP immunoprecipitation (GFP:IP). Samples were analysed by immunoblotting. Data represent three independent biological repeats.

3. The experiment shown in Fig 3G is key as it demonstrates that Leu89 and Pro91 of SteE are essential for inducing GSK3b Tyr kinase activity. However the experiment is missing controls in which GSK3b was left out of the reaction. This is important to ensure the observed kinase activity was not due to contamination from SteE preparations isolated from 293 cells (I realize these were GSK3a/b knockout cells, but it doesn't rule out other Tyr kinases co-purifying with SteE).

We apologise for this oversight. In our 2020 manuscript we extensively demonstrated that GSK3 was the kinase mediating direct phosphorylation, as compared to a contaminating kinase. In the revised manuscript we have included the requested control, demonstrating there is no phosphorylation in the absence of GSK3 and added this as **new Figure EV4B**, and copied below for ease.

Figure EV4B: SteE L89 is required for tyrosine phosphorylation mediated by GSK3. The indicated GFP-tagged proteins were immobilised on beads after expression from *GSK3 α/β ^{-/-}* 293ET cells. Tyrosine phosphorylation by GSK3 β was analysed in an *in vitro* kinase reaction containing 1 mM ATP, 0.2 μ M His₆-STAT3¹²⁷⁻⁷¹⁵, with or without recombinant Avi-GSK3 β ^{S9A} (0.2 μ M).

4. Fig 4B showing the AlphaFold multimer model would benefit from zoomed in views illustrating the presumed interaction interface, which is not well-represented in the current version of this panel. The main text needs to elaborate on the model. There is a generic statement that it is consistent with the HDX-MS data, but no details are given, and aside from a few highlighted residues, it's not clear what residues form the interface (like the helix whose binding site spans the GSK3b N- and C-lobes). The ipTM score plots for the AlphaFold multimer model in Fig 4B should also be shown to allow others to assess the confidence in the model, and it would be useful to include a PDB file. In the model, GSK3b would not be Tyr phosphorylated - does it capture the catalytic site conformation in reported crystal structures?

We have now carried out new structure predictions of the SteE-GSK3 complex using AlphaFold3 and provide the iPTM/pTM scores as requested, as well as the PAE and pLDDT scores per atom (**new Figure EV5A-B**). Given the low confidence of the prediction (apart from the similarity with Axin described below) we do not think that providing zoomed in views or a detailed discussion of the predicted interface would be of much value.

We have modelled the complex without and with Tyr216 phosphorylation, which did not make a significant difference to the final outcome. This is not surprising given that crystal structures of non-phosphorylated and phosphorylated GSK3 are very similar.

5. The AlphaFold model predicts multiple points of interaction beyond the short motifs identified by sequence alignment, but the validity of this model was not tested. For example, the similarity of the binding mode of a predicted helical region of SteE in the model to that of the GSK3b scaffold Axin is interesting. In Axin this is an amphipathic helix that makes extensive hydrophobic contacts with the kinase. Are the hydrophobic residues conserved to SteE? Does mutation of these residues disrupt the interaction with and Tyr phosphorylation of GSK3b?

As described in our answer to point 4 we have now repeated structure predictions for SteE (**Figure EV1B**) and the GSK3 β -SteE (**Figure 4B-C**) complex using AlphaFold3. This highlighted that overall confidence levels of the predicted complex structure are low, apart from region S98-S118, which overlaps well with the HDX-MS data presented and has, in part, already been analysed by mutagenesis as shown in Figure 2A. The hydrophobic residues in the Axin peptide that mediate the interaction with GSK3 β are indeed conserved in SteE, which is now shown in **new Figure EV5C**. However, we don't believe that further mutagenesis beyond that presented in this manuscript would provide any additional insight at this point, given the low confidence of the predicted complex structure.

Minor comments:

1. Can the authors speculate as to why Thr91 is required for SteE to interact with GSK3b? The authors suggest that it must be phosphorylated, but is it possible that non-phosphorylated Thr91 will also interact? A T91V substitution or dephosphorylating SteE with phosphatase treatment could address this question.

In Figure EV 1E, and summarised in our model in Figure 4, we show that SteE and GSK3 form an initial complex *in vitro* without phosphorylation. To test whether phosphorylation of T91, or the residue itself, or a combination of both, then mediates GSK3 interaction requires the analysis of complex formation using phospho-locked peptides. This is beyond the scope of this manuscript but is reflected in our discussion on lines 386-388.

2. HDX-MS panels in Fig 1 don't have X-axis labels. I gather each datapoint is a distinct peptide species shown in Fig S2.

Apologies for not including the x-axis label. This is the "Residue number", which has now been added.

Dear Dr. Thurston,

Thank you for the submission of your revised manuscript to our editorial offices. I have now received the reports from the three referees that I asked to re-evaluate the study, you will find below. As you will see, the referees now fully support the publication of the study in EMBO reports. Referee #1 has two remaining points I ask you to address in a final revised manuscript.

- We plan to publish your manuscript as Report, as there are only 4 main figures and 5 EV figures. For a Scientific Report we require that results and discussion sections are combined in a single chapter called "Results & Discussion". Please do this for your manuscript. For more details please refer to our guide to authors:
<http://www.embopress.org/page/journal/14693178/authorguide#researcharticleguide>
- Please also order the manuscript sections like this, using (only) these names:
Title page - Abstract - Keywords - Introduction - Results & Discussion - Methods - Data availability section - Acknowledgements - Disclosure and Competing Interests Statement - References - Figure legends - Expanded View Figure legends
- Please remove the information on Supplementary materials and Appendix Tables S1-S3 from the end of the manuscript.
- The Data availability section (DAS) is restricted for information on primary datasets produced in the study (e.g. RNA-seq, ChIP-seq, structural and array data, mass spec data) that are deposited in a public database and deposited source data. Thus, please remove all further text not related to externally deposited datasets from this section (regarding lead contact, materials availability and codes). Moreover, please remove now the referee access information from the DAS and make sure the dataset is public latest upon publication of the paper. Please also add a direct link to the dataset.
- The nomenclature for EV figure files and legends wrong. Please use Figure EVx for file names, legends and callouts.
- Please check again that the number "n" for how many independent experiments were performed, their nature (biological versus technical replicates), the bars and error bars (e.g. SEM, SD) and the test used to calculate p-values is indicated in the respective figure legends (main and EV figures). Please also check that all the p-values are explained in the legend, and that these fit to those shown in the figure. Please provide statistical testing where applicable. Please avoid the phrase 'independent experiment' but clearly state if these were biological or technical replicates. Please also indicate (e.g. with n.s.) if testing was performed, but the differences are not significant. In case n=2, please show the data as separate datapoints without error bars and statistics. See also:
<http://www.embopress.org/page/journal/14693178/authorguide#statisticalanalysis>

If $n < 5$, please show single datapoints for diagrams. Moreover:

- Please note that the exact p values are not provided in the legends of figures 4A.
- Please note that information related to n is missing in the legend of figure 4A

In addition, I would need from you uploaded separately:

Best,

Referee #1:

I previously reviewed this manuscript at EMBO J, and having read the rebuttal and re-read the new version, think it is improved,

answers my minor points and is ideal for EMBO R. Two minor things.

1) What is Kinase Buffer (Cell Signaling)? I couldn't find this in the manuscript, but assume it must contain Mg²⁺ ions, but important for follow-up studies that this is defined.

2) Thanks for deposition of proteomics data via ProteomeXchange

Referee #2:

With the added data and changes to the text, the authors have addressed my previous concerns. I would like to commend the authors on a very nice study.

Referee #3:

As I'd indicated in my original review, this manuscript makes an important contribution to understanding how the bacterial virulence factor SteE co-opts host signaling pathways through the kinase GSK3 β . The new experiments included are of high quality and have addressed all of my concerns; in particular I am more convinced that SteE induces GSK3 β Tyr kinase activity specifically. While I do not doubt the importance of the SteE L/x-G-x-P motif in this process, I am still not entirely convinced that this represents eukaryotic motif mimicry. However, the authors are appropriately cautious in this interpretation and I am satisfied that we can "agree to disagree" on this point.

All editorial and formatting issues were resolved by the authors.

Teresa Thurston
University of Oxford
Sir William Dunn School of Pathology
South Parks Road
Oxford OX1 3RE
United Kingdom

Dear Dr. Thurston,

I am very pleased to accept your manuscript for publication in the next available issue of EMBO reports. Thank you for your contribution to our journal.

Yours sincerely,
